# Erythropoietin promotes hippocampal mitochondrial function and enhances cognition in mice

Robert A. Jacobs [1,2,8], Mostafa A. Aboouf [1,3,4,8], Christina Koester-Hegmann[1,5], Paola Muttathukunnel[5,6], Sofien Laouafa[7], Christian Arias-Reyes [7], Markus Thiersch [1,3], Jorge Soliz[7], Max Gassmann [1,3] & Edith M. Schneider Gasser [1,5,6 ✉]

Erythropoietin (EPO) improves neuronal mitochondrial function and cognition in adults after brain injury and in those afflicted by psychiatric disorders. However, the influence of EPO on mitochondria and cognition during development remains unexplored. We previously observed that EPO stimulates hippocampal-specific neuronal maturation and synaptogenesis early in postnatal development in mice. Here we show that EPO promotes mitochondrial respiration in developing postnatal hippocampus by increasing mitochondrial content and enhancing cellular respiratory potential. Ultrastructurally, mitochondria profiles and total vesicle content were greater in presynaptic axon terminals, suggesting that EPO enhances oxidative metabolism and synaptic transmission capabilities. Behavioural tests of hippocampus-dependent memory at early adulthood, showed that EPO improves spatial and short-term memory. Collectively, we identify a role for EPO in the murine postnatal hippocampus by promoting mitochondrial function throughout early postnatal development, which corresponds to enhanced cognition by early adulthood.

[1] Institute of Veterinary Physiology, Vetsuisse-Faculty, University of Zurich, Zurich, Switzerland. [2] Department of Human Physiology & Nutrition, University of Colorado, Colorado Springs, CO, USA. [3] Zurich Center for Integrative Human Physiology (ZIPH), University of Zurich, Zurich, Switzerland. [4] Department of Biochemistry, Faculty of Pharmacy, Ain Shams University, Cairo, Egypt. [5] Institute of Pharmacology and Toxicology, University of Zurich, Zurich, Switzerland. [6] Center for Neuroscience Zurich (ZNZ), Zurich, Switzerland. [7] Faculty of Medicine, Centre Hospitalier Universitaire de Québec (CHUQ), Institut Universitaire de Cardiologie et de Pneumologie de Québec, Université Laval, Québec, QC, Canada. [8] These authors contributed equally: Robert A. Jacobs, Mostafa A. Aboouf. ✉email: edith.schneidergasser@uzh.ch

Erythropoietin (EPO) continues to develop as a promising neuroprotective agent with widespread clinical relevance[1,2] since the discovery that EPO and its receptor (EPOR) are expressed in the central nervous system (CNS)[3]. EPO can mediate neuroprotection following traumatic injury, hypoxia-ischemic insults, excitotoxicity, inflammatory brain injury[4,5], and retinal protection following hypoxia[6] or vascular injury[7,8]. EPO prevents hippocampal neurodegeneration and attendant cognitive impairment in rodents presenting with diabetes[9] or Alzheimer-like diseases[10,11] as well as reduces long-term spatial-memory deficits across various animal models of neonatal brain injury[12,13]. In humans, EPO has been shown to enhance memory retrieval by increasing hippocampal plasticity[4,14–17] and improving neurogenesis[10,14,18–20] in both healthy individuals and those affected by various neurodegenerative diseases and dementia. The neuroprotective potential of EPO also extends to neonatology, since it has been reported to reduce perinatal brain injury[21,22] and improve neurological outcomes in very prematurely born neonates[21,23,24].

EPO neuroprotection involves the activation of multiple downstream signaling pathways, including extracellular signal-regulated kinases 1 and 2 (Erk1/2) and phosphatidylinositol 3-kinase (PI3K)/protein kinase B (AKT)[13], which concomitantly upregulate anti-apoptotic, anti-inflammatory, anti-oxidative, and anti-cytotoxic pathways[13,21,22]. The cellular effects of these pathways (i.e., apoptosis, redox balance, cytotoxicity, and inflammation) all relate through their dependence on mitochondrial control. Indeed, EPO has been shown to enhance mitochondrial biogenesis through AKT signaling in cardiomyocytes[25,26], and activation of Erk1/2 promotes dynamin-related protein 1-dynamin 1 (DRP1-DNM1)-dependent mitochondrial fission and cell survival[27,28].

The importance of mitochondria in neuronal development and function is highlighted by their role in promoting neuronal differentiation, neurotransmission, and synaptic pruning[29]. The close relationship mitochondria share with neurodevelopment and cognition is also evidenced by the influence of mitochondrial morphology and function on synaptic vesicle pool regulation, spatial memory, and working memory throughout neuronal maturation[30,31]. EPO has been reported to support neuronal mitochondria and prevent memory impairment in animal models of multiple sclerosis, sleep deprivation, neurodegeneration, and brain injury[32–36]. Furthermore, EPO prevents neurobehavioral deficits in the hippocampus of young rats exposed to intermittent hypoxia[37] as well as a murine model of sleep apnea[38] by managing cellular oxidative stress. However, questions relating to the influence of EPO regulating brain mitochondria during development remain to be answered.

We have recently reported that EPO signals on CA1 pyramidal cells in the hippocampus and improves hippocampal postnatal neuronal maturation by reducing cell death and promoting synaptogenesis[39]. Putative mitochondrial mechanism(s) in promoting these processes are unexplored. Additionally, whether EPO improves hippocampal-mediated cognitive function in healthy animals remains unknown.

Accordingly, the aim of this research was to analyze the influence of EPO on brain mitochondria throughout postnatal development and examine whether postnatal EPO availability influences cognition in mice. This work demonstrates that EPO in the CNS activates Erk1/2 and AKT pathways in the hippocampus. Constitutive cerebral EPO overexpression and high-dose intraperitoneal (i.p.) administration of EPO during three consecutive days, coincides with increased mitochondria number, size, and respiration. EPO overexpression increases mitochondria number and function at postnatal ages (P) 14 to 21. The postnatal rise in mitochondria was also identified with electron microscopy (EM)

in presynaptic axon terminals, along with higher vesicle number, which suggests that EPO promotes neuronal mitochondrial function and enhances the putative reserve pool of the synaptic vesicles. Transcriptional signals controlling mitochondrial biogenesis and dynamics were altered with EPO-associated increases in peroxisome proliferator-activated receptor gamma coactivator 1-alpha (PGC1α), mitofusin 1 (MFN1), mitofusin 2 (MFN2), and DRP1-DNM1. Thus, EPO-mediated activation of Erk1/2 and AKT pathways throughout postnatal development corresponds with attendant improvements in mitochondrial function and hippocampal-specific cognition.

## Results

Respirometry, mRNA, and protein analyses in transgenic mice constitutively overexpressing recombinant human (rh) EPO specifically in the brain (Tg21) and wildtype (WT) control mice were done across postnatal ages (P): 3, 7, 14, 21, and 60 (Fig. 1a). EM ultrastructural analysis was done at P14, and behavioral analysis was performed during early adulthood (P45–60) in both genotypes (Fig. 1a).

**EPO overexpression in the CNS activates Erk1/2 and AKT pathways in the hippocampus.** Measurements of total hippocampal EPO expression was greater in Tg21 mice at all postnatal ages (2-way ANOVA, $F_{(1,91)} = 533$, $p < 0.0001$, Fig. 1b). Specifically, the average EPO protein expression in WT hippocampus was $38.7 \pm 3$ pg/mg immediately after birth (P1), which then decayed rapidly with postnatal development to $6.3 \pm 2$ pg/mg at adulthood. Alternatively, hippocampal EPO expression in Tg21 mice was 2.3-fold higher at P1 ($89.5 \pm 2$ pg/mg), 4.5-fold higher at P3 ($134.5 \pm 2$ pg/mg), 13-fold higher at P7 ($181.3 \pm 3$ pg/mg), 16-fold higher at P14 ($88.9 \pm 4$ pg/mg), and 10-fold higher at P21 ($66.4 \pm 3$ pg/mg) (Fig. 1b). Thereafter, EPO overexpression in Tg21 animals diminished to a 4-fold higher concentration ($22 \pm 6$ pg/mg) than WT in adulthood (P60). No differences were observed between genotypes regarding spleen weight (WT, $55.1 \pm 14$ mg vs. Tg21, $54.9 \pm 23$ mg; student's $t$-test, $p = 0.99$), hematocrit (WT, $39 \pm 7\%$ vs Tg21, $39.4 \pm 5\%$; student's $t$-test, $p = 0.88$), plasma hemoglobin (WT, $15.2 \pm 1$ vs Tg21, $16.2 \pm 1$; student's $t$-test, $p = 0.07$), and plasma EPO concentrations (WT, $45.1 \pm 23$ vs Tg21, $43.5 \pm 18$ pg/mL; student's $t$-test, $p = 0.88$) measured at P21 (Supplementary Fig. 1a). Murine EPO (mEPO) expression was hardly detected in the brain of WT and Tg21 mice while rhEPO was only detected in the brain of Tg21 mice (2-way ANOVA, $F_{(1,16)} = 275$, $p < 0.0001$; Supplementary Fig. 1b). Constitutive overexpression of rhEPO was restricted to the brain of Tg21 groups (P5 and P21) (1-way ANOVA brain, $F_{(2,9)} = 30.57$, $p < 0.0001$) with a minor transient elevation detected in the liver at P5 but no differences in kidney or spleen (Supplementary Fig. 1c).

Hippocampal EPOR mRNA expression, assessed by RT-PCR analysis, increased throughout postnatal development in both WT and Tg21 mice reaching a 4-fold increased expression at P60 compared to P3 (2-way ANOVA, $F_{(4,50)} = 53.11$, $p < 0.0001$; Fig. 1c) with no significant differences between genotypes (2-way ANOVA, $F_{(1,50)} = 1.51$, $p = 0.22$; Fig. 1c). EPOR mRNA expression was 8-fold higher in the spleen than hippocampus at P60 in both genotypes (2-way ANOVA, $F_{(1,18)} = 18.67$, $p = 0.0004$, Supplementary Fig. 1d), suggesting that EPO sensitivity in the hippocampus is lower than that of hemopoietic tissue. Cortical EPOR mRNA expression was negligible after birth and throughout development with no differences between genotypes, apart from a transient rise in WT mice at P14 (2-way ANOVA, $F_{(1,38)} = 2.77$, $p = 0.1$; Supplementary Fig. 1e). Thus, postnatal

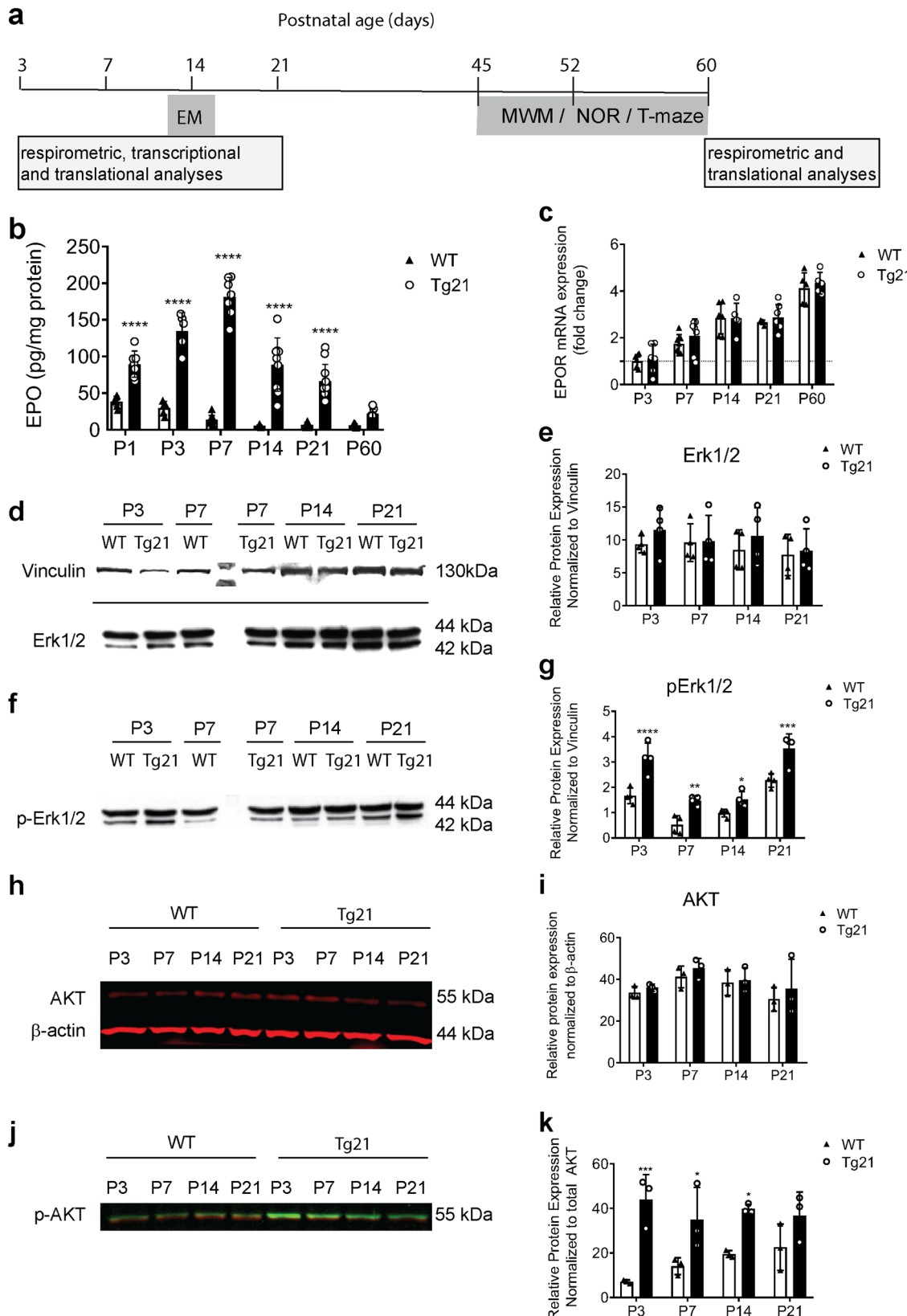

*EPOR* transcription in the brain is largely specific to the hippocampus, reaching a plateau towards adulthood (P21).

Intracellular EPO/EPOR signaling acts through multiple pathways including Erk1/2 and AKT, with both pathways involved in regulating mitochondrial biogenesis, mitochondrial dynamics, metabolic control, and cell fate[25,27]. Therefore, we analyzed the control of hippocampal EPO overexpression on Erk1/2 and AKT activation throughout postnatal development by western blot (Fig. 1d–k and Supplementary Fig. 3). Total Erk1/2 (Fig. 1d, e) and phosphorylated (p)Erk1/2 (Fig. 1f, g) band intensities were normalized to vinculin expression. There were no differences in total Erk1/2 protein between WT and Tg21 mice (2-way

**Fig. 1 Erythropoietin (EPO) overexpression in the central nervous system (CNS) activates Erk1/2 and AKT pathways in the hippocampus. a** Schematic diagram of the experimental design used for wildtype (WT) and transgenic (Tg21) animals that constitutively overexpress cerebral EPO. **b** Total (murine and recombinant human) EPO protein expression in the hippocampus across postnatal (P) ages: 1, 3, 7, 14, 21, and 60. EPO is greatly overexpressed in the hippocampus of Tg21 mice throughout postnatal development, decaying to its lowest difference compared to age-matched WT controls by adulthood (P60); 2-way ANOVA, $F_{(1,91)} = 533$, ****$p < 0.0001$. **c** *EPOR* mRNA expression in the hippocampus of WT and Tg21 mice across postnatal ages P: 3, 7, 14, 21, and 60, normalized to GAPDH. An age-induced increase in EPOR is shown with no difference between genotypes; 2-way ANOVA, $F_{(1,50)} = 1.51$, $p = 0.22$. **d** Representative uncropped western blot images of total Erk1/2 protein expression across postnatal ages P: 3, 7, 14, and 21 in WT and Tg21 mice with a vinculin loading control. **e** Quantification of total Erk1/2 protein shows no differences between genotypes; 2-way ANOVA, $F_{(1,24)} = 1.24$, $p = 0.28$. **f** Representative uncropped western blot images of phosphorylated (p)Erk1/2 protein expression across postnatal ages P: 3, 7, 14, and 21 in WT and Tg21 mice with vinculin load control. **g** Quantification of pErk1/2 shows highest levels of phosphorylation at P3 and P21 in both genotypes and 2-fold higher activation in the Tg21 mice across all measured ages; 2-way ANOVA, $F_{(1,24)} = 63.54$, ****$p < 0.0001$, with multiple comparison: *$p < 0.05$, **$p < 0.01$, ***$p < 0.001$, ****$p < 0.0001$. **h** Representative image of total AKT protein expression with β-actin loading control (red; top panel) in WT and Tg21 mice at postnatal ages P: 3, 7, 14, and 21. **i** Quantification of total AKT shows no change throughout development or difference between genotypes; 2-way ANOVA, $F_{(1,16)} = 1.35$, $p = 0.26$. **j** Representative image of pAKT (green) superimposed over total AKT (red) in WT and Tg21 mice at postnatal ages P: 3, 7, 14, and 21. **k** Quantification of pAKT shows the lowest levels of phosphorylation in WT mice at P3 with a gradual increase until P21, and Tg21 animals demonstrate similarly higher values at all ages; 2-way ANOVA, $F_{(1,16)} = 43.92$, ****$p < 0.0001$, with multiple comparisons: *$p < 0.05$, ***$p < 0.001$. Barplots with SD bars.

ANOVA, $F_{(1,24)} = 1.24$, $p = 0.28$; Fig. 1e). In contrast, pErk1/2 was at least 2-fold higher in Tg21 mice across all postnatal ages. The greatest difference in pErk1/2 between genotypes, identified primarily by the 42 kDa band, occurred at postnatal ages P3 and P21 (2-way ANOVA, $F_{(1,24)} = 63.54$, $p < 0.0001$, Fig. 1g). Fluorescent western blot analyses of total AKT (red, Fig. 1h and Supplementary Fig. 2) and pAKT (green, Fig. 1j and Supplementary Fig. 2) showed an 8-fold greater expression of pAKT over total AKT at P3, the 4-fold difference a P7, and a 2-fold difference at P14 and P21 in Tg21 mice (2-way ANOVA, $F_{(1,16)} = 43.92$, $p < 0.0001$; Fig. 1k) with no differences in total AKT between genotypes (2-way ANOVA, $F_{(1,16)} = 1.35$, $p = 0.26$; Fig. 1i). No significant changes in total Erk1/2 (2-way ANOVA, $F_{(3,24)} = 0.43$, $p = 0.73$, Fig. 1e) and in AKT (2-way ANOVA, $F_{(3,16)} = 2.72$, $p = 0.08$; Fig. 1i) were observed during development in either genotype. These data suggest that EPO-dependent activation of Erk1/2 and AKT pathways occurs in the hippocampus without influencing the total protein expression of these transcriptional regulators. pAKT did not vary in Tg21 or WT throughout development (2-way ANOVA, $F_{(3,16)} = 0.72$, $p = 0.55$; Fig. 1k). Moreover, pErk1/2 was higher at P3 and P21 when compared to P7 and P14 in both genotypes (2-way ANOVA, $F_{(3,16)} = 46.99$, $p < 0.0001$; Fig. 1g). Since activation of Erk1/2 and AKT are associated with mitochondrial metabolism and dynamics[25,27,28], we examined whether EPO-mediated activation of Erk1/2 and AKT influences mitochondrial morphology and function.

**EPO overexpression in the CNS promotes mitochondrial respiration in the postnatal hippocampus.** We used high-resolution respirometry to assess respiratory control in hippocampal tissue from both Tg21 and WT control mice with representative respirometric traces collected from P21 WT (top) and Tg21 (bottom) animals illustrated in Fig. 2a. Mass-specific NADH-linked respiration ($P_{CI}$; 2-way ANOVA, $F_{(1.66)} = 15.74$, $p = 0.002$; Fig. 2b) and non-coupled NADH- and succinate-linked respiration ($E_{CI+CII}$; 2-way ANOVA, $F_{(1,66)} = 22.22$, $p < 0.0001$; Fig. 2d) were higher at P14 and P21 in the Tg21 mice. Maximal rates of NADH- and succinate-linked respiration ($P_{CI+CII}$; $F_{(1,66)} = 9.21$, $p = 0.003$; Fig. 2c) were also higher in Tg21 mice vs WT controls at P21.

There was also an effect of age on mass-specific oxygen consumption rates (OCR), with an increase observed as age advanced in WT (1-way ANOVA, $F_{(4,35)} = 25.2$, $p < 0.0001$; Fig. 2e) and Tg21 (1-way ANOVA, $F_{(4,35)} = 24.0$, $p < 0.0001$; Fig. 2f) animals. In general, age-associated increases in hippocampal

OCR occurred earlier and more abruptly in the postnatal development of Tg21 mice between P7 and P14 (Fig. 2f) when compared to their WT counterparts that demonstrated a more gradual increase from P3 to P21 (Fig. 2e). Specific intra-genotypic differences in respiratory states across ages are shown in Fig. 2e, f for WT and Tg21 mice, respectively. Collectively, these data suggest that overexpression of cerebral EPO facilitates greater hippocampal respiratory capacity earlier in postnatal development.

Mass-specific respiration also increased with age throughout postnatal development in the cortex (WT: 1-way ANOVA, $F_{(2,14)} = 5.6$, $p = 0.02$, Tg21: 1-way ANOVA, $F_{(2,13)} = 10.15$, $p = 0.002$, Supplementary Fig. 3a) and in WT brainstem (1-way ANOVA, $F_{(2,18)} = 11.55$, $p = 0.0006$, Supplementary Fig. 3b) but no differences between genotypes were identified (2-way ANOVA, $F_{(1,30)} = 0.15$, $p = 0.7$, Supplementary Fig. 3a and 2-way ANOVA, $F_{(1,36)} = 0.0002$, $p = 0.99$, Supplementary Fig. 3b, respectively). This suggests that EPO-mediated influence over mitochondria is limited to the hippocampus.

**EPO overexpression in the CNS facilitates changes in respiratory control by increasing mitochondrial content in the hippocampus during postnatal development.** Cytochrome $c$ oxidase (COX; complex IV) activity, validated as a surrogate of mitochondrial content in human skeletal muscle[40], differed between genotypes throughout postnatal development with higher measures observed at P14, P21, and P60 in Tg21 animals compared to WT (2-way ANOVA, $F_{(1,66)} = 36.77$, $p < 0.0001$; Fig. 3a). Like respiratory control (Fig. 2), an age-dependent increase in COX activity was observed in both genotypes with the greatest increases in COX activity occurring between P7 and P14 in Tg21 and between P14 and P21 in WT mice (2-way ANOVA, $F_{(4,66)} = 45.79$, $p < 0.0001$; Fig. 3a). Mitochondria-specific respiratory analysis (mass-specific respiration normalized to a surrogate of mitochondrial content, e.g., COX activity) helps distinguish qualitative alterations in respiratory control from quantitative differences driven primarily by mitochondrial content. When normalizing mass-specific OCR to COX, respiratory differences between genotypes were lost (Fig. 3b–f). These data suggest that mass-specific OCR differences between genotypes are most likely attributable to quantitative disparities in mitochondrial content as opposed to qualitative variations in mitochondrial function.

To further test the differences in hippocampal mitochondria between genotypes, we examined the ratio of mitochondria DNA (mtDNA, represented by ND1 gene) normalized to nuclear DNA (nDNA, represented by β2M) by qPCR as well as the expression of various mitochondrial proteins with Western blots across

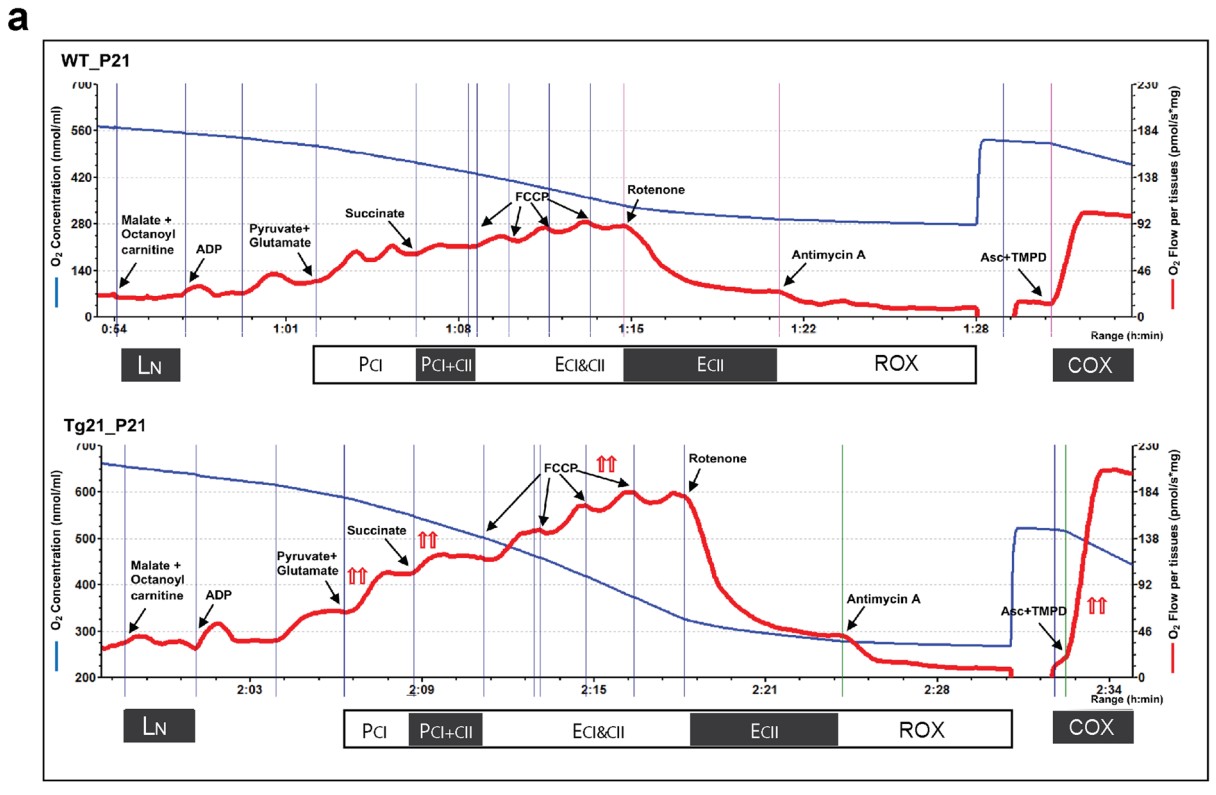

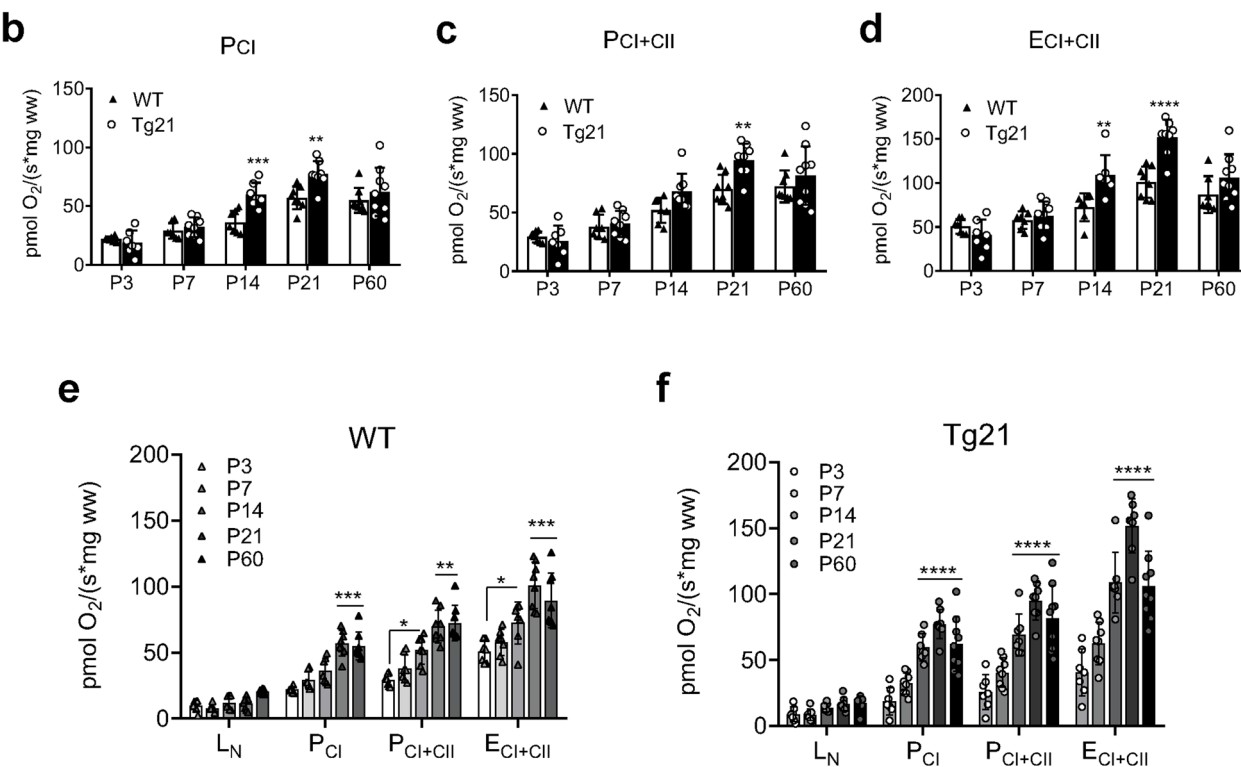

postnatal ages (Fig. 4). mtDNA/nDNA ratio was higher in Tg21 mice at P14 and P21 (2-way ANOVA, $F(1,18) = 26.19$, $p < 0.0001$; Fig. 4a). Furthermore, higher voltage-dependent anion channel 1 (VDAC1) expression, the most abundant protein of the outer membrane of mitochondria, was observed in Tg21 mice (Fig. 4b–c and Supplementary Fig. 4) with 4.9-fold higher protein

expression at P14 and 1.6-fold more protein at P21 compared to WT (2-way ANOVA, $F(1,16) = 73.94$, $p < 0.0001$; Fig. 4c). Mitochondrial proteins specific to the electron transport system were also higher in Tg21 mice compared to WT at various stages throughout postnatal development (Fig. 4d–i). Specifically, complex I (NDUFB8) was 2.7-fold higher at P7 and 4.3-fold

**Fig. 2 EPO overexpression in the CNS influences mass-specific respiratory control in the postnatal hippocampus. a** Representative respirometric traces from a WT (top panel) and a Tg21 (bottom panel) hippocampal tissue, which illustrates the change in oxygen concentration (nmol·ml$^{-1}$, left $y$ axis, blue line) and oxygen flux per mass (pmol $O_2$/s * mg ww, right $y$ axis, red line) in hippocampal tissue at P21. Respiratory states were achieved through the titration of various substrates, inhibitors, as well as a protonophore. The order of respiratory state analysis from beginning to end (left-to-right) with the respective substrates, inhibitors, or protonophores added, as fully explained in the methods, consisted of: leak without adenylates ($L_N$; addition of malate and octanoylcarnitine); coupled respiration with maximal electron input specific to mitochondrial complex I ($P_{CI}$; addition of ADP, pyruvate, and glutamate); maximal rates of coupled respiration with electron input specific to complex I and II ($P_{CI+CII}$; addition of succinate); maximal non-coupled respiration (E) with electron input from complex I and II ($E_{CI+CII}$; steps of carbonyl cyanide $p$-trifluoromethoxy phenylhydrazone, FCCP, addition until respiration ceases to increase); non-coupled respiration with maximal electron input specific to mitochondrial complex II ($E_{CII}$; addition of rotenone); and non-mitochondrial residual oxygen consumption (ROX; addition of antimycin). Following respiratory state analysis, ascorbate and $N,N,N',N'$-tetramethyl-1,4-benzenediamine, dihydrochloride (TMPD) were simultaneously added to assess cytochrome $c$ oxidase (COX; complex IV) activity. **b** Mass-specific $P_{CI}$ respiration in WT and Tg21 mice, showing higher oxygen consumption rates (OCR) at P14 and P21 in Tg21 mice; 2-way ANOVA, $F(1.66) = 15.74$, **$p = 0.002$. **c** Mass-specific $P_{CI+CII}$ respiration in WT and Tg21 mice, showing higher OCR at P21 in Tg21 mice; 2-way ANOVA, $F(1,66) = 9.21$, **$p = 0.003$. **d** Mass-specific $E_{CI+CII}$ in WT and Tg21 mice, showing higher OCR at P14 and P21 in Tg21 mice; 2-way ANOVA, $F(1,66) = 22.22$, ****$p < 0.0001$. **e** Mass-specific respiration for WT mice at postnatal ages of P: 3, 7, 14, 21, and 60. An age-dependent effect is observed in $P_{CI}$, $P_{CI+CII}$, and $E_{CI+CII}$; 1-way ANOVA, $F(4,35) = 25.2$, ****$p < 0.0001$. **f** Mass-specific respiration for Tg21 mice at postnatal ages of P: 3, 7, 14, 21, and 60. An age-dependent effect is observed in $P_{CI}$, $P_{CI+CII}$, and $E_{CI+CII}$; 1-way ANOVA, $F(4,35) = 24.0$, ****$p < 0.0001$, with multiple comparisons: **$p = 0.002$, ***$p < 0.001$, ****$p < 0.0001$. Barplots with SD bars.

higher at P14 (2-way ANOVA, $F(1,24) = 26.69$, $p < 0.0001$; Fig. 4e), complex II (SDHB) was 2.2-fold higher at P21 (2-way ANOVA, $F(1,24) = 10.93$, $p = 0.003$; Fig. 4f), complex III (UQCRC2) was 13.7-fold higher at P7 (2-way ANOVA, $F(1,24) = 62.26$, $p < 0.0001$; Fig. 4g), complex IV (COX-IV) was 6.9-fold higher at P7, 3.9-fold higher at P14, and 2-fold higher at both P21 and P60 (2-way ANOVA, $F(1,24) = 131.9$, $p < 0.0001$; Fig. 4h), and complex V (ATP5A) was 3.2-fold higher at P7 in Tg21 mice (2-way ANOVA, $F(1,24) = 15.24$, $p = 0.0007$; Fig. 4i).

**High-dose intraperitoneal (i.p.) administration of EPO increases hippocampal Erk1/2 and AKT activity as well as mitochondrial respiration and content.** To verify that hippocampal increases in pErk1/2, pAKT, respiratory rates, and mitochondrial content are not due to a non-specific effect of our transgenic (Tg21) model, WT mice received (50 μl) i.p. treatments of either saline or rhEPO (5 IU/g) for three consecutive days at the age of weaning (P23-25, Fig. 5a). This dose of EPO is high enough to facilitate transportation across the blood brain barrier[41]. At P26, twenty-four hours after the last i.p. injection, hippocampal Erk1/2 and AKT activation, respiratory control, and indices of mitochondrial content were assessed (Fig. 5c–l and Supplementary Fig. 5). Systemic administration of EPO resulted in the canonical increase in hematocrit from 38% to 50.4% (Student's $t$-test, $p < 0.0001$; Fig. 5b) that was not observed in saline-treated animals. While total AKT protein was higher in EPO-treated mice (Student's $t$-test, $p = 0.049$; Fig. 5d), there was no difference in Erk1/2 expression between EPO and saline-treated groups (Student's $t$-test, $p = 0.26$; Fig. 5e). pAKT and pErk1/2 were higher in EPO-treated mice (Student's $t$-test; $p = 0.03$, Fig. 5h and $p = 0.01$, Fig. 5i, respectively). EPO-treated mice expressed more mitochondrial protein including VDAC1 expression (Student's $t$-test, $p = 0.004$; Fig. 5f) and COX activity (Student's $t$-test, $p = 0.0006$; Fig. 5j) as well as mass-specific $P_{CI}$, $P_{CI+CII}$, and $E_{CI+CII}$ OCR (2-way ANOVA, $F(1,56) = 40.27$, $p < 0.0001$; Fig. 5k). When normalizing mass-specific respiration to COX, the differences between treatments were lost (2-way ANOVA, $F(1,56) = 0.77$, $p = 0.38$; Fig. 5l), further confirming that EPO increases mitochondrial content in the hippocampus. Collectively, these results are consistent with the effects of cerebral over-expression of EPO in the Tg21 line.

**EPO overexpression in CNS upregulates PGC1-1α, influences mitochondria dynamics, and ultimately increases mitochondria and vesicle number in presynaptic terminals.** Transcriptional expression of peroxisome proliferator-activated receptor-gamma coactivator 1α ($PGC1α$) throughout postnatal development (P3, P7, P14, P21) was assessed via RT-PCR in WT and Tg21 mice. An age-dependent increase in $PGC1α$ mRNA was evident in both genotypes with a 2.2-fold increase at P14-P21 compared to P3-P7 (2-way ANOVA, $F(3,18) = 46.93$, $p < 0.0001$; Fig. 6a), which generally corresponds with the age-dependent increases in hippocampal respiration (Fig. 2e, f). Moreover, transcriptional expression of $PGC1α$ was higher in Tg21 compared to WT at P21 (Student's $t$-test, P21, $p = 0.03$; Fig. 6a). Transcriptional expression of autophagy-related gene 5 ($ATG5$) showed an age-dependent increase in expression without any difference between genotypes (2-way ANOVA, $F(3,24) = 11.98$, $p = 0.001$; Fig. 6b).

Transcriptional control of mitochondrial pro-fusion, mitofusin 1 ($MFN1$) and mitofusin 2 ($MFN2$), as well as pro-fission, $DRP1$-$DNM1$ and dynamin 2 ($DNM2$), signaling was assessed via RT-PCR throughout postnatal development in WT and Tg21 mice. A 2.6-fold age-dependent increase in $MFN2$ mRNA was observed in WT and Tg21 at P14 and P21 over P3 and P7 (2-way ANOVA, $F(3,24) = 48.59$, $p < 0.0001$; Fig. 6c) with greater expression in Tg21 vs WT animals at P21 (2-way ANOVA, $F(1,24) = 7.2$, $p = 0.01$; Fig. 6c). $MFN1$ mRNA did not change with age but Tg21 animals demonstrated greater expression at P21 (2-way ANOVA, $F(1,24) = 1.143$, $p = 0.01$; Fig. 6d). No age-dependent changes or genotype-specific differences in $DNM2$ mRNA were observed (2-way ANOVA, $F(1,24) = 0.37$, $p = 0.54$; Fig. 6e). While transcriptional expression of $DRP1$-$DNM1$ also did not show age-dependent changes in either genotype, higher mRNA was identified in Tg21 vs WT mice at P21 (2-way ANOVA, $F(1,24) = 8.834$, $p = 0.0066$; Fig. 6f). These data suggest that EPO upregulates transcriptional signaling for mitochondrial biogenesis via $PGC1α$ and alters mitochondrial dynamics in the hippocampus.

Mitochondrial profiles in the hippocampus, along with quantitative analysis of presynaptic mitochondria content and vesicle number, were examined by electron microscopy (EM; Fig. 6g–k). The upregulation of hippocampal mitochondria as suggested by COX activity (Fig. 3a), mtDNA/nDNA (Fig. 4a), mitochondrial protein (Figs. 4 and 5), and $PGC1α$ mRNA (Fig. 6a) was confirmed via EM as evidenced by a higher number of mitochondria structures (Fig. 6g, arrows and Fig. 6h; Student's $t$-test, $p = 0.036$) as well as greater size (Student's $t$-test, $p = 0.002$; Fig. 6i). Importantly, total mitochondrial mass and vesicle number were both higher in presynaptic axon terminals of Tg21 mice irrespective of the synapse type (symmetric or asymmetric) (Fig. 6g insets, Student's $t$-test, $p = 0.0043$, Fig. 6j; and Student's $t$-test, $p < 0001$, Fig. 6k). These latter results suggest

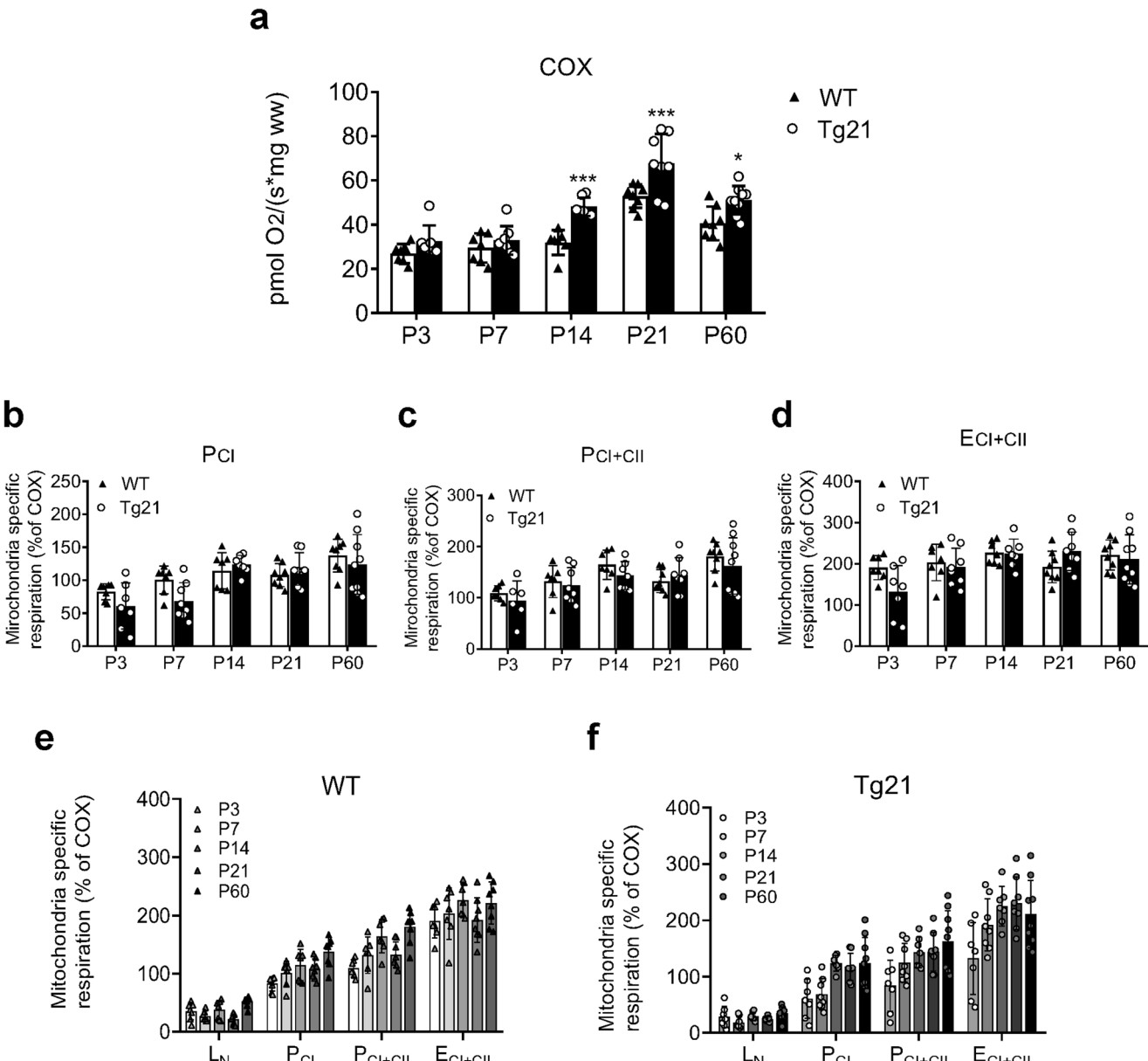

**Fig. 3 EPO overexpression in the CNS influences respiratory control by increasing mitochondria in the postnatal hippocampus. a** Cytochrome C oxidase activity (COX, pmol $O_2$/s * mg ww) in the hippocampus of WT and Tg21 throughout postnatal development. Tg21 mice show higher COX activity at P14, P21, and P60; 2-way ANOVA, $F(1,66) = 36.77$, ****$p < 0.0001$, with multiple comparisons: *$p < 0.05$, and ***$p < 0.001$. Mitochondria-specific respiration for **b** $P_{CI}$; **c** $P_{CI+CII}$; and **d** $E_{CI+CII}$ throughout postnatal development in WT vs Tg21 mice. An increase in respiration throughout postnatal development is observed along with an increase in mitochondrial content in both genotypes for all respiratory states. No differences between genotypes are observed: (2-way ANOVA; $P_{CI}$, $F(1,66) = 2.40$, $p = 0.13$); $P_{CI+CII}$, $F(1,66) = 2.28$, $p = 0.13$; and $E_{CI+CII}$, $F(1,66) = 0.73$, $p = 0.40$. Mass-specific oxygen consumption rates (OCR) relative to COX (%) in **e** WT mice and **f** Tg21 mice for leak respiration without adenylates ($L_N$); complex I ($P_{CI}$); complex I and II ($P_{CI+CII}$); and maximal non-coupled respiration with electron input from complex I and II ($E_{CI+CII}$) at postnatal ages of P: 3, 7, 14, 21, and 60. Mass-specific respiratory differences between genotypes are lost when normalizing to COX, a surrogate of mitochondrial content; 1-way ANOVA: WT, $F(4,15) = 0.24$, $p = 0.91$; and Tg21, $F(4,15) = 0.45$, $p = 0.76$. Barplots with SD bars.

a possible increase in synaptic transmission influencing hippocampal function in Tg21 mice.

**EPO overexpression in CNS enhances hippocampal-dependent spatial navigation and short-term memory**
*Morris water maze (MWM) test.* Young adult mice from both genotypes successfully learned to find the hidden platform during the acquisition phase, as demonstrated by the escape latency and swimming path length (Fig. 7a, upper panels). A shorter escape latency was observed in Tg21 mice beginning on day 2 (d2) (2-

way ANOVA, $F(1,22) = 28.18$, $p < 0.001$; Fig. 7a upper left panel). The swimming path length was not different between genotypes during the acquisition phase (2-way ANOVA, $F(1,22) = 0.85$, $p = 0.36$; Fig. 7a, upper middle panel). However, the indication of enhanced learning in Tg21 compared to WT controls was evident across days with Tg21 mice showing longer swim paths on d1, negligible differences in swim paths on d2, and shorter swim paths on d3 and d4. Additionally, Tg21 mice exhibited faster swim speeds than WT controls across all days (2-way ANOVA, $F(1,22) = 62.4$, $p < 0.0001$; Fig. 7a upper right

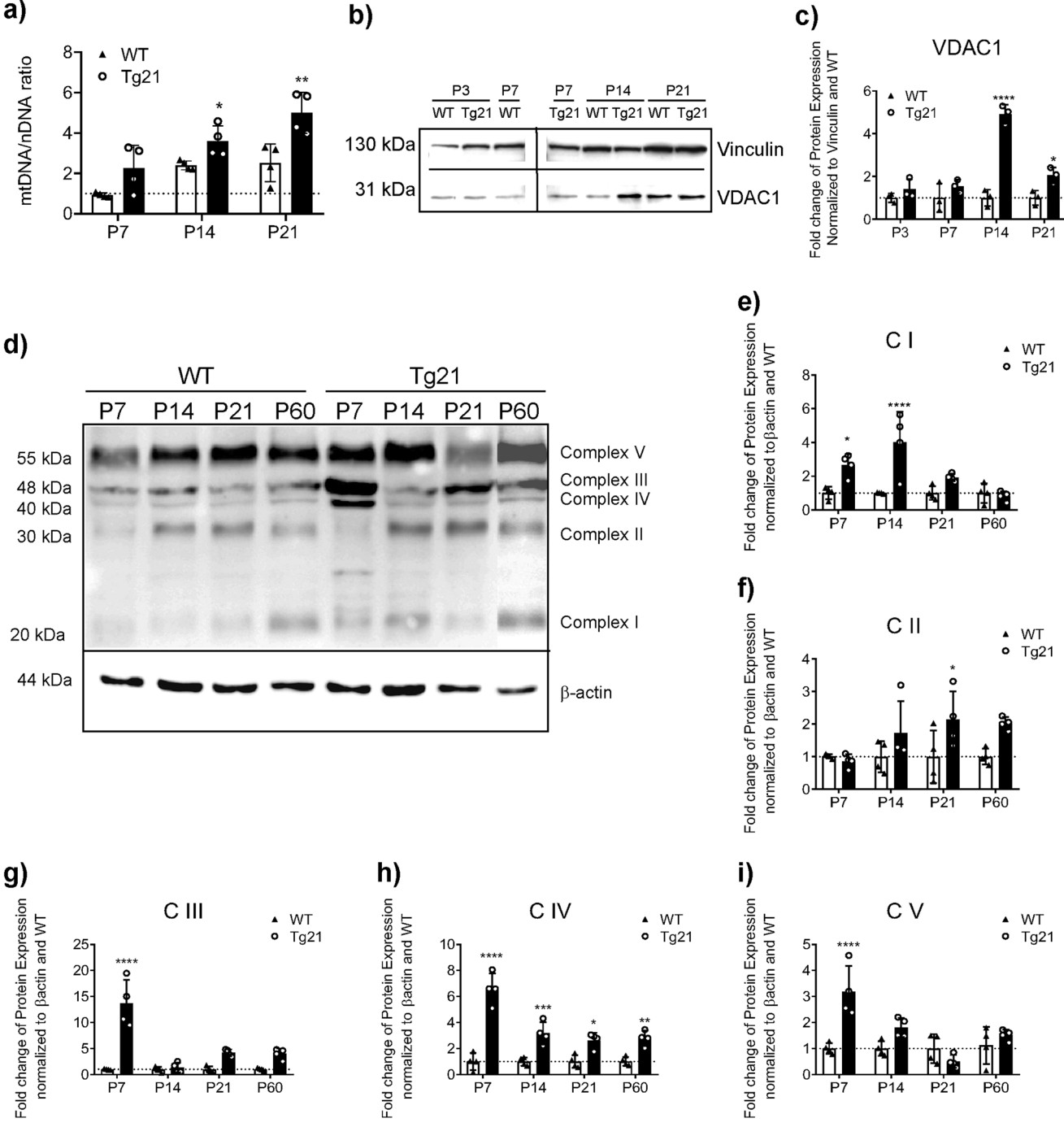

**Fig. 4 EPO overexpression in the CNS increases mitochondrial DNA, voltage-dependent anion channel 1 (VDAC1), and electron transport system proteins in the hippocampus throughout postnatal development. a** Fold change in mitochondrial (mt) DNA over nuclear (n) DNA ratio in WT and Tg21 mice across postnatal ages P: 7, 14, and 21 shows higher mtDNA to nDNA in Tg21 mice; 2-way ANOVA, $F_{(1,18)} = 26.19$, ****$p < 0.0001$, with multiple comparisons: *$p < 0.05$ and **$p < 0.01$. **b** Representative images of total VDAC1 with vinculin as a loading control across postnatal ages P: 3, 7, 14, and 21. **c** Quantification of VDAC1 protein expression normalized to vinculin and to WT at P3; 2-way ANOVA, $F_{(1,16)} = 73.94$, ****$p < 0.0001$. **d** Representative uncropped images of mitochondrial electron transport system protein expression and against β-actin as loading control in WT and Tg21 mice across postnatal ages P: 7, 14, 21, and 60. **e** Quantification of protein expression for mitochondrial complex I (CI), NADH dehydrogenase subunit NDUFB8; 2-way ANOVA, $F_{(1,24)} = 26.69$, ****$p < 0.0001$; **f** mitochondrial complex II (CII), succinate dehydrogenase complex iron-sulfur subunit B (SDHB); 2-way ANOVA, $F_{(1,24)} = 10.93$, **$p = 0.003$; **g** mitochondrial complex III (CIII), ubiquinol–cytochrome $c$ reductase core protein 2 (UQCRC2); 2-way ANOVA, $F_{(1,24)} = 62.26$, ****$p < 0.0001$; **h** mitochondrial complex IV (CIV), cytochrome $c$ oxidase (COX), subunit 1; 2-way ANOVA, $F_{(1,24)} = 131.9$, ****$p < 0.0001$; and **i** mitochondrial complex V (CV), ATP synthase alpha subunit (ATP5A); 2-way ANOVA, $F_{(1,24)} = 15.24$, ***$p = 0.0007$. Mitochondrial protein expression is higher in Tg21 mice at P7 (CIII, CIV, and CV), P14 (VDAC1, CI, and CIV), P21 (VDAC1, CII, and CIV), and P60 (CIV). Dotted lines represent values of 1. Multiple comparisons: 2-way ANOVA: *$p < 0.05$, **$p < 0.01$, ***$p < 0.001$, and ****$p < 0.0001$. Barplots with SD bars.

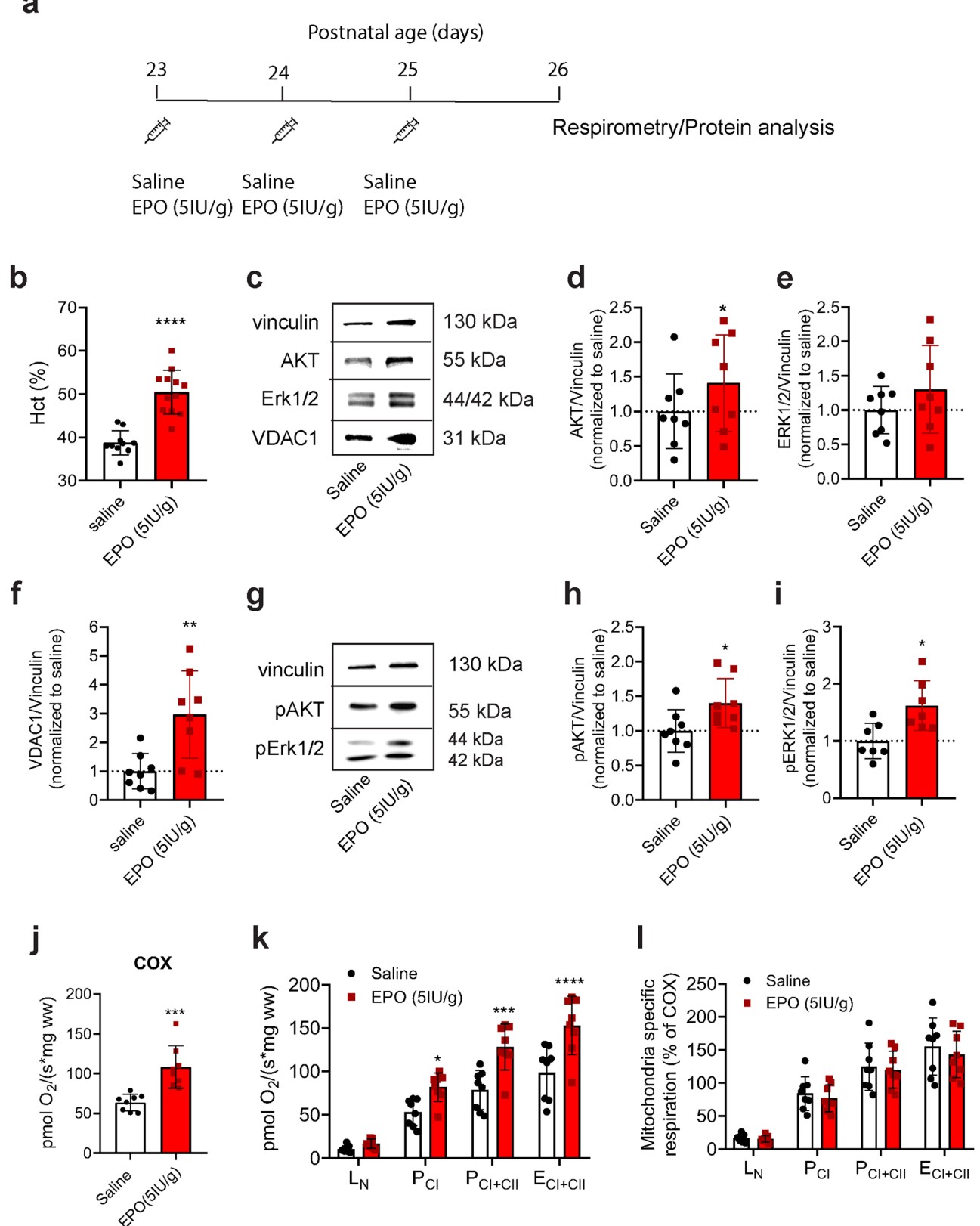

panel). During the reversal trial (d5) (Fig. 7a lower panels), both groups showed an increased preference for the training quadrant, and this preference was stronger in Tg21 animals (2-way ANOVA, $F(1,22) = 10.5$, $p < 0.0001$). Additionally, Tg21 mice presented shorter escape latency (2-way ANOVA, $F(1,22) = 10.8$, $p < 0.001$; Fig. 7a lower left panel), shorter swim path length (2-way ANOVA, $F(1,22) = 12.1$, $p < 0.01$; Fig. 7a lower middle panel), and faster swim speed (2-way ANOVA, $F(1,22) = 18.56$, $p < 0.0001$; Fig. 7a lower right panel) when finding the new hidden platform location. These results suggest that Tg21 mice have improved spatial learning and memory compared to WT controls.

**Fig. 5 Intraperitoneal (i.p.) EPO injections in WT mice resulted in greater pAKT, pErk1/2, higher rates of respiration, and more mitochondrial content in the hippocampus. a** Schematic diagram of EPO (red bars) or saline (white bars) treatment in mice. Male animals were injected for three consecutive days after weaning (P23 to P25) with either recombinant human (rh)EPO (5 IU/g) or saline solution. At P26 hippocampal tissue was taken for respirometry and protein analysis. **b** Changes in hematocrit (%) after treatment; unpaired $t$-test, ****$p < 0.0001$. **c** Representative images of total AKT, total Erk1/2, and voltage-dependent anion channel 1 (VDAC1) protein expression with vinculin as a loading control. **d** Quantification of AKT normalized to vinculin and saline; unpaired $t$-test, *$p = 0.049$. **e** Quantification of Erk1/2 protein expression normalized to vinculin and saline; unpaired $t$-test, $p = 0.26$. **f** Quantification of VDAC1 protein expression normalized to vinculin and saline; unpaired $t$-test, **$p < 0.004$. **g** Representative images of total phosphorylated pAKT and pErk1/2 protein expression with vinculin as a loading control. **h** Quantification of pAKT normalized to vinculin and saline; unpaired $t$-test, *$p = 0.03$. **i** Quantification of pErk1/2 normalized to vinculin and saline; unpaired $t$-test, *$p = 0.012$. **j** Cytochrome $C$ oxidase activity (COX, pmol $O_2$/s * mg ww) in the hippocampus of WT mice randomly treated with either saline or EPO (5 IU/g). EPO-treated mice show higher COX activity; unpaired $t$-test ***$p = 0.0006$. **k** Hippocampal mass-specific oxygen consumption rates (OCR) from WT control (saline) and EPO (5 IU/ml) injected mice. $P_{CI}$ respiration, $P_{CI+CII}$ respiration, and $E_{CI+CII}$ respiration is increased in EPO-treated mice; 2-way ANOVA, $F_{(1,56)} = 40.27$, ****$p < 0.0001$, with multiple comparisons: *$p < 0.05$, ***$p < 0.001$, and ****$p < 0.0001$. **l** Mitochondria-specific OCR (mass-specific OCR normalized to COX activity, a surrogate for mitochondrial content). No differences in mitochondria-specific respiration are observed between saline and EPO-treated mice; 2-way ANOVA, $F_{(1,56)} = 0.77$, $p = 0.38$. Barplots with SD bars.

*Novel object recognition (NOR)*. The cognitive impact of EPO on recognition memory was assessed via novel object recognition (NOR) tasks (Fig. 7b). Both Tg21 and WT mice were habituated to learn the location and become familiar with two equal objects over a single 5 min learning event. Subsequently, short-term and long-term memory was examined when testing the recognition of a novel object after 1 and 24 h inter-trial intervals (ITI), respectively (Fig. 7b, upper panel). During the habituation phase, Tg21 animals explored the empty arena more intensively than WT mice resulting in more total locomotor activity ($4.96 \pm 6.8$ vs. $3.97 \pm 6$ m/min, respectively, Student's $t$-test, $p < 0.001$). During the training phase, total object exploration time was similar between genotypes. In the short-term memory test, both WT and Tg21 mice spent more time exploring the novel object. However, Tg21 mice exhibited a stronger preference for the novel object reflected by an object discrimination index (equation (1)) above 0.5. Long-term memory was then tested 24 h later when the mice were given 5 min in the arena with a novel displaced object. In this task, both genotypes equally recognized the novel object with a similar object discrimination ratio (equation (1)) of $0.28 \pm 0.18$ for WT and $0.25 + 0.2$ for Tg21 (2-way ANOVA, $F_{(1,68)} = 6.18$, $p = 0.01$; Fig. 7b, lower panel). Although this test is not exclusively hippocampal, as visual cues are involved in task analysis, it is a valid test to assess cognition in young adult mice with the speed in learning locations highlighting cognitive flexibility.

*T-maze spatial working memory*. Impact of EPO on spatial working memory showed spontaneous alternation to be more than 75% correct in both genotypes (Fig. 7c). Nevertheless, Tg21 mice were faster in learning the choice ($10.13 \pm 0.74$ s for WT and $7.37 \pm 0.70$ s for Tg21; Student's $t$-test, $p = 0.0004$), and had less errors than WT in the second and third test of each. When the ITI was 24 h, however, no difference between genotypes was observed (t4–6; Student's $t$-test, $p = 0.4$; Fig. 7c, bottom panel) Thus, only an increase in short-term spatial working memory was observed in Tg21 mice.

Collective, cognitive testing demonstrates that EPO overexpression improves performance speed, hippocampal-mediated spatial learning and memory, and short-term memory while long-term memory remained unaltered.

## Discussion

The aim of this research was to analyze the influence of EPO on brain mitochondria throughout early postnatal development and examine whether EPO availability improves cognition in mice. Our primary findings reveal that EPO promotes postnatal increases in hippocampal mitochondria, respiratory potential, and enhances cognition. More specifically, elevated cerebral EPO during early postnatal development increases hippocampal pErk1/2 and pAKT while hastening local mitochondrial maturation by approximately one week (~33% increase), which functionally results in greater oxidative potential. Expedited mitochondrial development in the hippocampus is also evident in presynaptic terminals along with greater vesicle numbers, both of which coincide with improved spatial and short-term memory at early adulthood. Collectively, these findings add to the clinical non-hematopoietic relevance of EPO on stimulating neural development and cognition (see Fig. 8 for results summary).

EPO/EPOR signaling is dependent upon the concentration of EPO and expression of EPOR. Expression of EPOR transcripts increases throughout postnatal development in the hippocampus, mainly in CA1 pyramidal neurons, with the greatest expression achieved at weaning (P21)[39]. This observation was confirmed in the current study, as *EPOR* mRNA increased throughout postnatal development up to P21. Alternatively, postnatal *EPOR* mRNA expression in cortical areas was mostly negligible. This implies a role of EPO in the postnatal maturation of CA1 principal neurons during the time of development when these cells undergo dendritic arborization, axon growth, spine formation, synaptogenesis[42], and when GABAergic synapses form within the postsynaptic populations of CA1 neurons[43]. A transient hippocampal hyperexcitability in the second postnatal week has been reported as a result of axonal remodeling[44]. Hyperexcitability causes brief episodes of local hypoxia in CA1 pyramidal neurons[45]. Therefore, it is conceivable that these transient episodes of local hypoxia trigger EPO and EPOR production at this age, enhancing synaptogenesis, circuit formation, and, consequently, cognition. Evidence of a local hypoxic influence on the expression of both EPO and *EPOR* in pyramidal cells, as well as its impact in spine density, has been previously demonstrated[46]. *EPOR* expression in developing embryonic mouse brain is similar to adult hematopoietic tissue[47], as similarly observed in humans[48]. However, high embryonic *EPOR* expression in the brain decreases ~20-fold over 10 days leading up to birth[49]. Here we show in adult mice that transcriptional *EPOR* expression in the brain is lower than the spleen by approximately eight orders of magnitude. Since *EPOR* expression closely relates to cellular EPO sensitivity, the relatively high concentrations of EPO in the brain achieved in this study by means of transgene or high-dose EPO injections are suggested to be essential for promoting mitochondrial development in the hippocampus of mice.

Intracellular EPO/EPOR signaling acts through multiple pathways including Erk1/2, AKT, and Janus kinase/signal transducers and activators of transcription (JAK/STAT). Analysis in this study focused on AKT and Erk1/2 pathways, as they are more

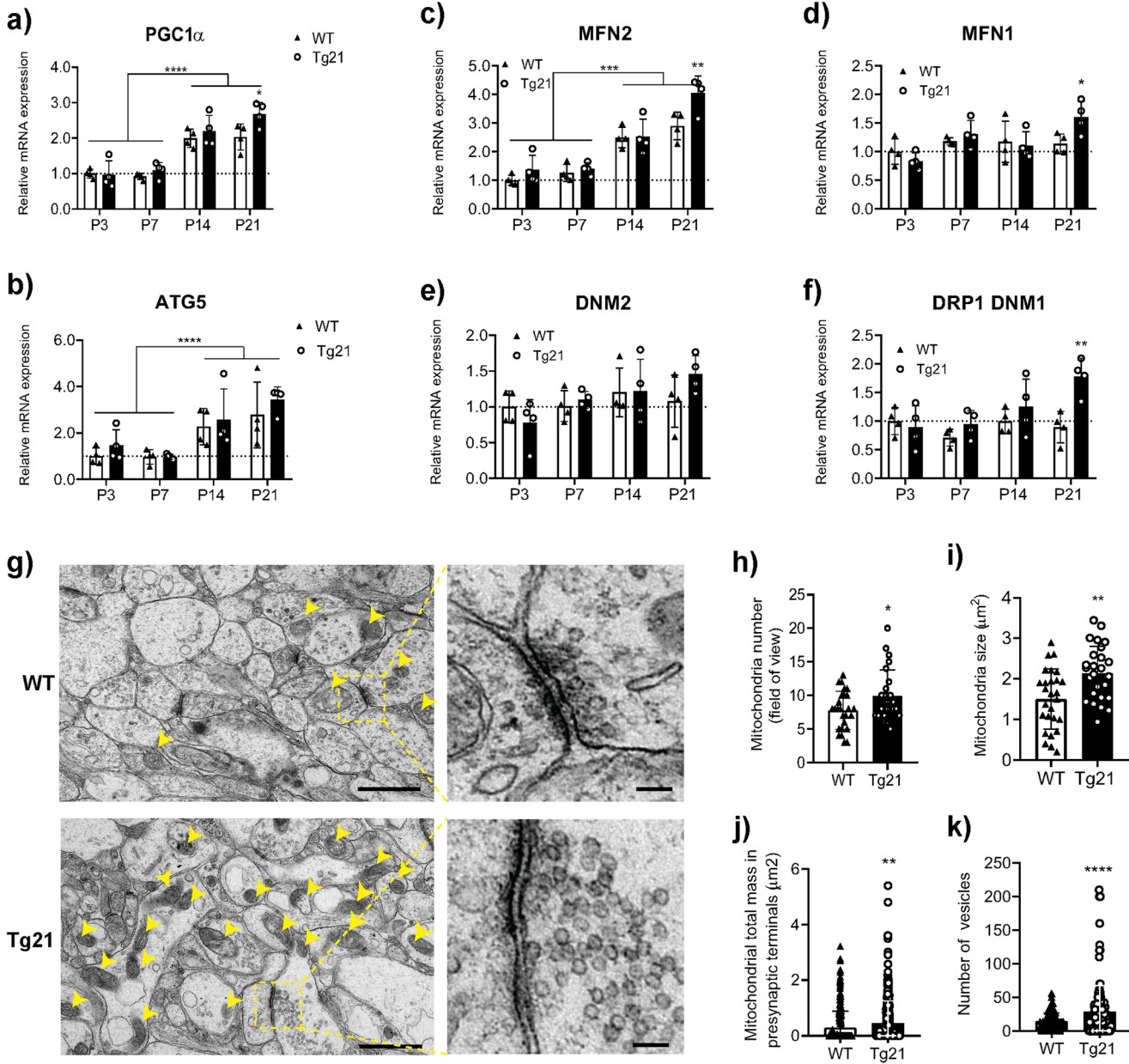

**Fig. 6 EPO overexpression in CNS upregulates *PGC1-1α*, alters control of mitochondria dynamics, increases mitochondria size, increases mitochondria number, and increases the number of vesicles in presynaptic terminals of the hippocampus. a** *PGC-1α* mRNA expression in hippocampus from WT and Tg21 mice. *PGC1α* increases with age in both genotypes; 2-way ANOVA, $F_{(3,24)} = 46.93$, ****$p < 0.0001$. At P21, a higher expression is observed in Tg21 mice; LSD post hoc, *$p = 0.03$. **b** *ATG5* mRNA expression in hippocampus from WT and Tg21 mice. *ATG5* demonstrates age-dependent increases with no difference between genotypes; 2-way ANOVA, $F_{(3,24)} = 11.98$, ****$p = 0.0001$. mRNA expression of pro-fusion: **c** Mitofusin 2 (*MFN2*) and **d** mitofusin 1 (*MFN1*), and of pro-fission: **e** dynamin 2 (*DNM2*), and **f** dynamin-related protein 1-dynamin 1 (*DRP1-DNM1*) in WT and Tg21 hippocampus. Transcriptional control of fusion (both *MFN1* and *MFN2*) is enhanced in Tg21 mice at P21; 2-way ANOVA: *MFN1*, $F_{(1,24)} = 1,143$, *$p = 0.30$; and *MFN2*, $F_{(1,24)} = 7.216$, **$p = 0.01$. An age-dependent increase in *MFN2* is observed in both genotypes; 2-way ANOVA, $F_{(3,24)} = 48.59$, ***$p < 0.001$. Pro-fission *DNM2* shows no change in expression across development nor between genotypes; 2-way ANOVA, $F_{(1,24)} = 0.37$, $p = 0.54$. Pro-fission *DRP1-DNM1* is higher in Tg21 at P21; 2-way ANOVA, $F_{(1,24)} = 8.834$, **$p = 0.007$. *p*. **a–f** Relative mRNA expression levels of the aforementioned genes are quantified by qPCR and normalized to *ACTB*. **g** Representative electron microscope (EM) images of hippocampal CA1 area from WT and Tg21 mice at P14. More mitochondrial profiles (yellow arrows) and a higher number of vesicles in presynaptic axon terminals are observed in Tg21 mice. Scale bar: 1 µm (right panels) and 100 nm (panel insets). **h** Quantification of mitochondria number in the total field of view; unpaired *t*-test, *$p = 0.036$. **i** Mitochondria size; unpaired *t*-test, $p = 0.002$. **j** Mitochondria total mass in presynaptic terminals; unpaired *t*-test, **$p = 0.0043$. **k** Vesicle numbers in presynaptic terminals; unpaired *t*-test, ****$p < 0001$. Barplots with SD bars.

commonly reported as key regulating pathways in mitochondrial biogenesis, mitochondrial dynamics, and metabolic control[25–28]. Here we show that constitutive overexpression of cerebral EPO, as well as high-dose i.p EPO injections shown to traverse the blood–brain barrier[41], increased pErk1/2 and pAKT levels in the hippocampus with negligible changes in total respective protein expression. During the early postnatal period, ERK participates in the maturation process of dendritic trees and synaptogenesis in

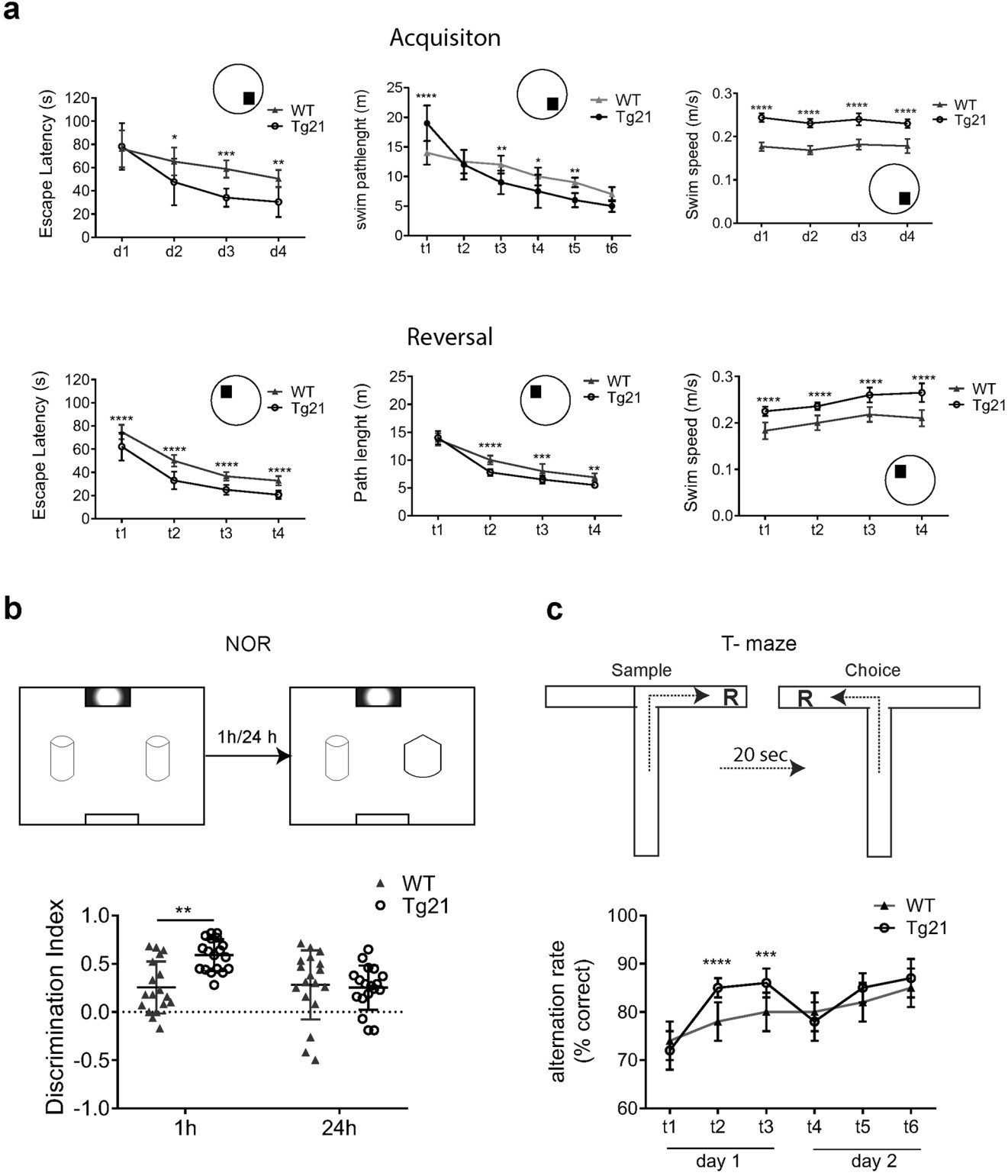

the hippocampus[50]. During neural differentiation, Erk1/2 signaling is associated with increased transcriptional expression of *PGC-1α*[51]. However, Erk1/2 generally promotes mitochondrial fission via DRP1 phosphorylation of S616 and suppressing MFN1 via phosphorylation of T562[27,28] under pro-apoptotic conditions[52]. Alternatively, AKT promotes a hyperfused mitochondrial network by suppressing DRP1 with reciprocal phosphorylation of DRP1 at S637 and reducing phosphorylation of S616 via proto-oncogene serine/threonine-protein kinase (Pim-

1)[53]. EPO-activated AKT increases mitochondrial biogenesis through eNOS by which an increase in nitric oxide (NO) upregulates PGC-1α, as observed in cardiomyocytes[25,26]. The mechanistic/mammalian target of rapamycin complex 1 (mTORC1) has also been shown to upregulate mitochondria and increase respiration via inhibition of the eukaryotic translation initiation factor 4E (eIF4E)-binding proteins (4EBPs)[54], and both Erk1/2 and AKT signaling promote the activation of mTORC1. Thus, EPO-EPOR signaling through Erk1/2 and AKT pathways

**Fig. 7 EPO enhances hippocampal-dependent spatial navigation and improves short-term memory. a** Morris water maze (MWM) tests was performed in WT and Tg21 mice at early adulthood (P48-60). Outcome variables relating to the acquisition phase (upper panels) and reversal phase (lower panels) are shown. Circular insets show the position of the hidden platform in the water maze. Escape latency (s) is quicker (2-way ANOVA, $F(1,22) = 28.18$, ***$p < 0.001$), swim path length (m) is equal (2-way ANOVA, $F(1,22) = 0.85$, $p = 0.36$), and swim speed (m/s) is faster (2-way ANOVA, $F(1,22) = 62.4$, ****$p < 0.0001$) in Tg21 mice during acquisition. During reversal, time in the training quadrant was higher in Tg21 mice; 2-way ANOVA, $F(1,22) = 105$, ****$p < 0.0001$. Tg21 animals demonstrated shorter scape latency (s), 2-way ANOVA, $F(1,22) = 10.8$, ***$p < 0.001$, shorter swim path length (m), 2-way ANOVA, $F(1,22) = 12.1$, **$p < 0.01$, and faster swim speed (m/s), 2-way ANOVA, $F(1,22) = 18.6$, ****$p < 0.0001$ than WT controls during reversal. Multiple comparison: *$p < 0.05$, **$p < 0.01$, ***$p < 0.001$, and ****$p < 0.0001$. **b** Representative diagram of the Novel Object Recognition (NOR) test, as fully explained in the methods. Object recognition relating to short-term memory function using 1 h inter-trial-intervals (ITI) and long-term memory function using 24 h ITI is shown with object discrimination index. Tg21 animals demonstrated longer total exploration time and an improved discrimination index when testing short-term memory. Animals from both genotypes appeared to recognize the replaced object equally when testing long-term memory; 2-way ANOVA, $F(1,68) = 6.18$, **$p = 0.01$. **c** T-maze spatial working-memory test shows spontaneous alternation above 75% in WT and Tg21 mice. Tg21 mice make less errors than WT at the second and third test from each day reflecting a better short-term but no improvement in long-term memory as observed in t4 after 24 h; 2-way ANOVA, $F(1.132) = 14.17$, ***$p = 0.0003$. Multiple comparisons, ***$p < 0.001$, and ****$p < 0.0001$. Barplots with SD bars.

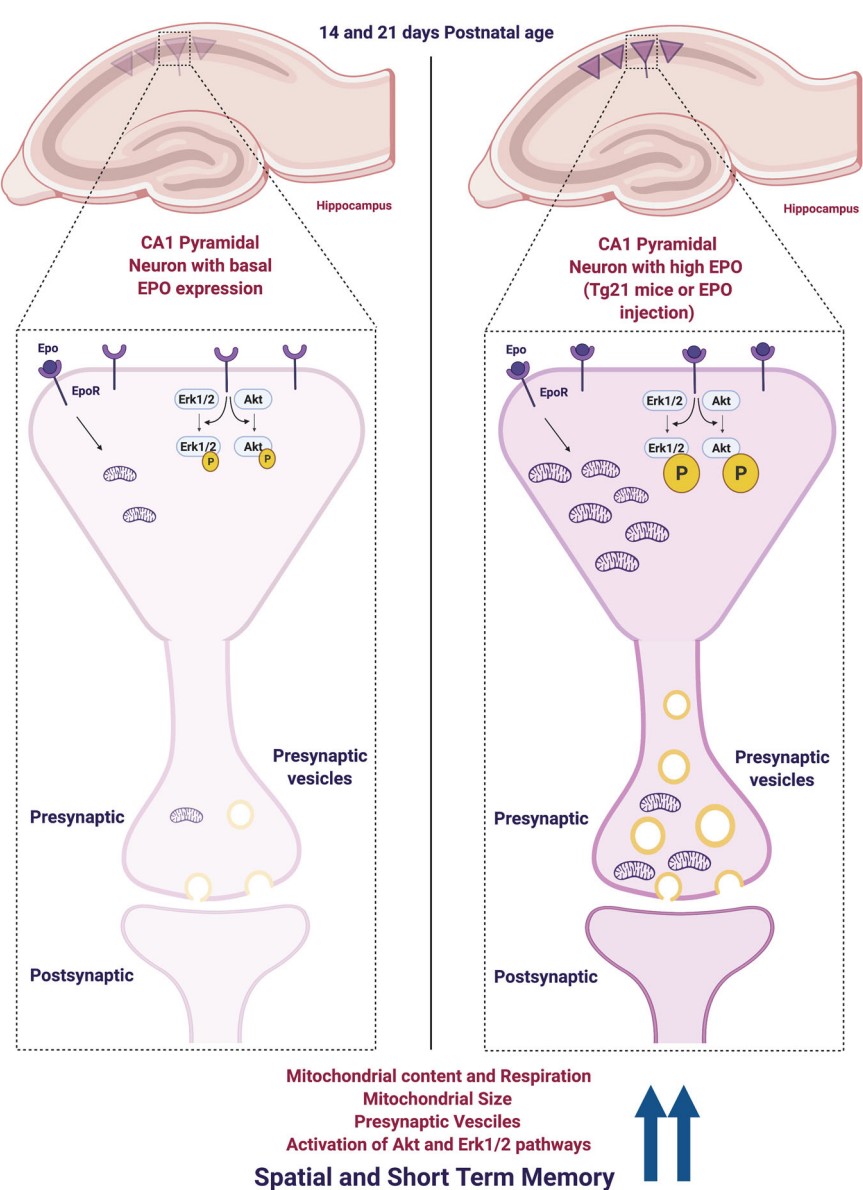

**Fig. 8 Illustrative summary of proposed EPO functions in hippocampal CA1 pyramidal cells at postnatal ages P14 and P21.** Hippocampal CA1 pyramidal cells are represented by triangles. When EPO-EPOR binding increases (e.g., Tg21 animals or WT mice treated with high doses of EPO), increased phosphorylation of Erk1/2 and AKT precedes an increase in mitochondria content, size, and function. More mitochondria are also observed in presynaptic terminals, along with an increase in vesicle numbers. As a result, animals with greater hippocampal respiratory potential show enhanced spatial learning and short-term memory at adulthood. (Figure created with BioRender.com).

could increase hippocampal mitochondria and respiratory potential.

High-resolution respirometry analyses demonstrated higher OCR for 3 different respiratory states ($P_{CI}$, $P_{CI+CII}$, and $E_{CI+CII}$) during postnatal development (P14 and/or P21) in Tg21 mice as well as P26 WT mice treated with EPO when compared to their respective control groups. Additionally, COX activity, ratio of mtDNA to nDNA, an assortment of mitochondrial proteins, as well as mitochondrial number and size assessed via electron microscopy were all greater with EPO. The higher respiratory potential with EPO appears to result from quantitative mitochondrial increases/expansion and not qualitative alterations in respiratory control, as differences in mass-specific OCR were lost when normalizing respiration to a surrogate of mitochondrial content (e.g., COX activity).

Transcriptional promotion of mitochondrial biogenesis was also higher in Tg21 animals as evidenced by more hippocampal *PGC-1α* mRNA at P21. Astrocyte differentiation and synaptogenesis have been shown to be influenced by *PGC-1α* in the postnatal prefrontal cortex[55]. We have also previously shown EPO-facilitated improvements in synaptogenesis during postnatal hippocampal development[39]. Thus, EPO-directed P*GC-1α* upregulation may influence synaptogenesis in the hippocampus. Transcriptional control of mitochondrial dynamics in the hippocampus was also altered with EPO, as pro-fusion (*MNF1* and *MNF2*) and pro-fission (*DRP1-DNM1*) signals were higher at P21. Recent evidence identified distinct DRP1-mediated signaling profiles that result in mitochondrial degradation or biogenesis in which the origin of fission differs[56]. Midzone mitochondrial fission promotes mitochondrial proliferation whereas peripheral mitochondrial fission sequesters dysfunctional components of the organelle for mitophagic control. Although we are unable to verify the specific fission profiles or origins of fission in the current study, peripheral fission is preceded by evidence of mitochondrial dysfunction (i.e., oxidative stress, drop in membrane potential, and/or loss of proton motive force) as well as lysosomal-mitochondrial contact[56]. Respirometric analyses in the current study suggest no difference in ROS production ($L_N$ state respiration) as well as greater membrane potential and proton motive force in Tg21 and EPO-treated WT mice ($P_{CI}$, $P_{CI+CII}$, and $E_{CI+CII}$ respiration). Also, we found no indication of the difference in lysosomal activation as evidenced by the transcriptional consistency of *ATG5* between genotypes throughout postnatal development. Given these findings, along with the observations that mitochondrial number and size were greater in Tg21 vs WT controls, EPO appears to increase mitochondrial biogenesis in the hippocampus early in life.

Mitochondrial plasticity is a key factor in metabolic adaptation, and alterations in shape and content can reflect changes in respiratory potential. Greater mitochondrial content was observed specifically in presynaptic terminals in combination with increased vesicle numbers at P14 in Tg21 mice. Although total vesicle number in synaptic terminals were counted and reported, not just reflecting those in contact with the active zone, synaptic vesicles are recruited to the active zone to discharge neurotransmitters during intense stimulation. This process is dependent on mitochondrially derived ATP[57]. Therefore, higher OCR and mitochondrial content in presynaptic neurons suggest a higher energetic potential to support greater vesicle release. A greater number of glutamatergic and GABAergic synapses and increased inhibitory postsynaptic current amplitude reported at P14 in Tg21 mice[39] also support the concept that increased respiratory potential is coupled with more synaptogenesis during postnatal development, and these EPO-facilitated differences may enhance cognition.

Mitochondrial dysfunction leads to severe defects in hippocampal-dependent cognitive function, such as memory and learning[58], associated with several neurodevelopmental disorders and degenerative diseases[59,60]. EPO has been shown to be an effective drug to restore mitochondrial dysfunction and cognition after brain traumatic injury[61] and in neurodegeneration[11,62–65]. EPO treatment improves neuronal mitochondrial activity and upregulates brain hemoglobin expression in mouse models of multiple sclerosis (MS)[35]. It is therefore likely that EPO overexpression causes an increase in neural hemoglobin during development stimulating neuronal mitochondrial function. Our paired observations of concentrated EPOR expression on CA1 cells[39] and higher rates of cellular respiration in the hippocampus of Tg21 mice led us to speculate that EPO overexpression may also improve hippocampal-mediated cognition.

It is well documented that EPO administration or overexpression enhances spatial memory across several models of disease[30,35,36,66]. Hippocampal CA1 principal cells have a well-defined role as being cognition-relevant but hypoxia-vulnerable[67]. Mitochondrial function, especially as it pertains to the hippocampus, has also been shown to influence spatial memory throughout life[68,69]. Spatial memory and short-term memory were tested using the MWM, NOR, and T-maze spatial working-memory tests. The MWM test showed significantly shorter escape latency during acquisition and reversal phases with reduced swim path length. These measures were all influence by the genotypic difference in task performance speed. We previously reported improved exercise performance in Tg21 mice with a longer time to exhaustion that was independent of systemic hematological differences (e.g., hematocrit)[70]. Perhaps the faster swim speed observed in the current study could be partially explained by the diminished influence of central fatigue on physical function in Tg21 animals, as metabolic control in the brain plays a critical role in limiting maximal exercise capacity[71]. While higher mitochondrial content was observed in Tg21 mice earlier in life (P14-21), maximal coupled respiration, $P_{CI+CII}$, the respiratory state that best reflects in vivo oxidative phosphorylation potential, did not differ between WT controls and Tg21 mice at P60. Moreover, the shorter swim path length of Tg21 mice to find the platform, suggests improved spatial orientation most likely attributed to enhanced plasticity as shown previously[13,16,72]. The NOR test with spatial cues examined the influence of EPO on short-term and long-term memory. Throughout the training phase, Tg21 mice showed increased activity, but total exploration time was like WT controls. Ultimately, our findings suggest that EPO overexpression may improve short-term memory, but long-term memory remains uninfluenced. The role of the hippocampus, cerebral cortex, and their integrated communication in developing short- and long-term memory is not fully understood. However, the neural mechanisms used to establish short- and long-term memory are known to be somewhat independent[73]. While we cannot currently provide a clear description explaining the disparate influence of EPO on short- and long-term memory, our observations suggest a role for EPO to enhance short-term memory. Finally, T-maze spatial working memory showed no differences in total spontaneous alternation between genotypes, but Tg21 mice were faster in the choice and made fewer errors in the second and third test from each day, further reflecting improved learning and short-term memory.

In this study, we provide insights on the mechanisms by which EPO may enhance cognition in early life. We show that EPO signaling through Erk1/2 and AKT corresponds with improvements in respiratory potential and mitochondrial content, such as the greater number and size of mitochondria identified in presynaptic terminals of the hippocampus, which are observable by the second postnatal week of development. This enhanced

energetic potential could support greater synaptogenesis and provide a cognitive advantage later in life. Indeed, constitutive expression of cerebral EPO also improved spatial and short-term memory in early adulthood. These findings highlight a therapeutical potential of EPO to stimulate hippocampal metabolism during maturation, which may improve neurodevelopment and cognition in, for example, premature children afflicted with hippocampal dysfunction following neonatal hypoxic-ischemic brain injury.

## Methods

**Animals**. Mitochondrial analyses were conducted with both sexes using WT controls and transgenic Tg21[74,75], mice at postnatal (P) ages: 3, 7, 14, 21, and 60 ($n = 8$). Behavioral tests were performed in early adulthood (P45–60). Tg21 hemizygous mice were backcrossed with C57Bl/6 mice for more than ten generations to obtain the corresponding control mice and Tg21 mice were bred to homozygosity.

All mice were bred and housed under standard conditions at the University of Zurich. Animals were kept on a 12/12-h light/dark cycle with food and water provided ad libitum. Animal experiments were performed following the ARRIVE guidelines and were approved by the Cantonal Veterinary Office of Zurich, Switzerland (ZH177_16).

**EPO intraperitoneal injections**. Male mice received i.p. injections with 5 IU/g rhEPO (NeoRecormon, Roche) or placebo (saline solution) for three consecutive days. EPO/saline treatment was initiated in P23 animals. Animals were sacrificed at P26, 24 h after the last treatment followed by respirometry or tissue collection for later protein analyses.

**Blood samples for measures of hematocrit, hemoglobin, and plasma EPO**. Mice were anesthetized with a subcutaneous injection containing 100 mg/kg ketamine (Ketasol-100; Dr. Graub, Switzerland), 20 mg/kg xylazine (Rompun; Bayer, Germany), and 3 mg/kg acepromazine (Sedalin, Switzerland). Blood samples were taken by cardiac puncture with a 27 G needle attached to a 1 ml heparinized syringe. The blood was immediately transferred into an Eppendorf tube (1 ml) to determine hematocrit, hemoglobin, and plasma EPO concentrations.

Duplicate measures of hematocrit were collected using microcapillary tubes (Micro haematocrit tube 100, Assistant) and immediately centrifuged (Hettich, Tuttlingen, Germany) for 5 min at $10^4$ rpm (Autokit II, Pharmap, Geneva, Switzerland). Hemoglobin concentration was determined using Abbott Cell Dyn 3500 (Abbott Diagnostic Division, Santa Clara, CA, USA). For assessing EPO in plasma, blood was collected in pre-cooled EDTA tubes, immediately spun for 10 min at $1500 \times g$ in a cooled centrifuge, plasma was collected and transferred to dry ice. All plasma samples were kept at $-80\,°C$ until assayed. Plasma EPO concentrations were assessed using a commercial radioimmunoassay (RIA) kit (Epo-Trac$^{TM 125}$I RIA kit; DiaSorin, USA). A known concentration of EPO antibody is incubated with 125-I radiolabelled antigens for binding. Blood plasma is added and unlabeled EPO antigens from plasma compete with the radiolabelled antigens for antibody binding sites. The higher the concentration of the EPO plasma antigens, more binds to the antibodies displacing more free radiolabelled antigens to the media. Radioactivity of the media is measured using a gamma counter (Quanta Smart for Tri-Carb 4910TR, PerkinElmer).

**Measurement of mouse EPO (mEPO) and recombinant human EPO (rhEPO) in brain, kidney, liver, and spleen tissues**. Concentrations of rhEPO were determined from frozen samples of brain, kidney, liver, and spleen from WT (P5) and Tg21 (both P5 and P21) mice using a commercial kit (Quantikine IVD ELISA, human erythropoietin, R&D Systems). Fresh wet weights of brain and spleen samples were registered for comparison.

mEPO and rhEPO levels in the brain were additionally assessed at P60 in WT and Tg21 mice using commercial kits (Quantikine ELISA, mouse erythropoietin, and human erythropoietin, R&D systems).

**Measurement of hippocampal EPO protein expression**. Hippocampal EPO from WT and Tg21 mice was assessed in snap-frozen samples throughout development using a radioimmunoassay (RIA) kit (Epo-Trac 1251 RIA kit; DiaSorin, Saluggia, Italy) as previously described[76].

**DNA extraction and mitochondrial copy number**. To extract genomic and mitochondrial DNA, 5–10 mg hippocampal tissue was digested by 100 µg/ml proteinase K in a lysis buffer (50 mM KCl, 10 mM Tris-HCl pH 8.3, 10 µg/ml gelatin, 0.045% Nonidet P-40, 0.045% Tween 20) at 55 °C overnight. After heat inactivation of proteinase K and centrifugation to remove cellular debris, DNA was precipitated by sodium acetate and ethanol. After centrifugation, DNA pellets were air-dried then resuspended in nuclease-free water. DNA concentration was analyzed using Nanodrop 2000 (ThermoFisher Scientific). The amount of

**Table 1 Primers used for RT-qPCR and annealing temperatures.**

| EPOR | F: 5′-ACAAGGGTAACTTCCAGCTGTG-3′<br>R: 5′-GATCCTCAGGGAAGGAGCTG-3′ | 65.6 °C |
|---|---|---|
| GAPDH | F: 5′-AATGTGTCCGTCGTGGATCTG-3′<br>R: 5′-CATACTTGGCAGGTTTCTCCAG-3 | 64.9 °C |
| MT-ND1 | F: 5′-AGTCTATGAGTTCCCCTACCA-3′<br>R: 5′-GTGAGTATTTGGAGTTTGAGGC-3′ | 63 °C |
| N-B2M | F: 5′-CGGCCTGTATGCTATCCAGA-3′<br>R: 5′-TCCACCCTGTAGCCTCAAAG-3′ | 65.3 °C |
| DNMI | F: 5′-AGGCATTACAAGGAGCCAGTCAAA-3′<br>R: 5′-GCAGCAGGTTCAAGTCA CAAAG-3′ | 66.1 °C |
| DNM2 | F: 5′-AGAGAATGAGGATGGAGCACAAGAG-3′<br>R: 5′-TTTGGCATAAGGTCACGGATGGA-3′ | 67.7 °C |
| MFN1 | F: 5′-AGCAAGGATGTAACCACTACTAAACA-3′<br>R: 5′-CTAAAGAGAGAGAGCAGGAAAGCA-3′ | 65.8 °C |
| MFN2 | F: 5′-TAAAGGATGTAGCAGGAGGAATGG-3′<br>R: 5′-TGTAGCAAGGCAGGGATGAG-3′ | 65.8 °C |
| PGC1a | F: 5′-TGGGTGTGCGTGTGTGTATGT-3′<br>R: 5′-GTCGCCCTTGTTCGTTCTGTTC-3′ | 67.6 °C |
| ATG5 | F 5′-TGTGCTTCGAGATGTGTGGTT -3′<br>R 5′- ACCAACGTCAAATAGCTGACTC-3 | 64.5 °C |
| ACTB | F: 5′-TTTCCAGCCTTCCTTCTTGGG-3′<br>R: 5′-GAGGTCTTTACGGATGTCAACG-3′ | 64.7 °C |

mitochondrial DNA (mtDNA) in tissue was estimated by the ratio of the mito-chondrial *MT-ND1* gene copy number and the nuclear *N-B2M* gene copy number[77,78]. Primers against both genes (Table 1) and SYBR Green (ThermoFisher Scientific, #A25741) were used for a semi-quantitative analysis by quantitative real-time PCR (7500 Fast Real-Time PCR System, ThermoFisher Scientific). The ratio of genomic (*N-B2M* gene) and mitochondrial (*MT-ND1*) DNA was determined by the ΔΔCt method[79].

**RNA extraction and mRNA expression analyses**. Five-to-ten mg of hippocampal and cortical tissues from both genotypes (WT and Tg21) and sexes (male and female) brain tissue at P3, P7, P14, and P21 were used to extract RNA using ReliaPrep RNA Tissue Miniprep System (Promega, #Z6110). Purity was determined spectrophotometrically (Nanodrop 2000, ThermoScientific). First-strand cDNA was obtained by reverse transcription using RevertAid First Strand cDNA Synthesis Kit (ThermoFisher Scientific, #K1622). Samples (5 ng/µl cDNA) were analyzed by a SYBR Green (ThermoFisher Scientific, #A25741) semi-quantitative real-time PCR (qRT-PCR) (7500 Fast Real-Time PCR System, ThermoFisher Scientific). Primers for mRNA expression analyses were designed with Primer 3.0. Software to amplify either human or murine genes without cross-specificity (Table 1). Oligo properties were calculated using Oligo Analyzer 3.1. Before using primers for mRNA expression analyses, they were validated by qRT-PCR by (i) melting curve analyses (mode integrated in the 7500 Fast Real-Time PCR System) as well as on (ii) acrylamide gels to confirm the size and purity of PCR products. mRNA expression levels were calculated using the ΔΔCt method[79] and normalized to either *ACTB* or *GAPDH*. Each group of samples was normalized to WT P3 hippocampus. Annealing temperatures are shown in Table 1.

**Western blot analysis**. Brain tissue samples from both sexes were collected from WT and Tg21 mice at P3, P7, P14, P21, and P60. Animals were deeply anesthetized with an i.p. injection of sodium pentobarbital (50 mg/kg; Nembutal, Kantonsapotheke Zürich) followed by decapitation and dissection of brain tissue on ice. Hippocampus and caudal cortex were dissected and placed on ice. Samples from each animal were independently processed ($n = 4$ animals/group). Total hippocampal tissue was homogenized by transferring the tissue through a 21 G needle 5 times in ice-cold 300 µl RIPA buffer (50 mM Tris/HCl pH 8.0, 150 mM NaCl, 1% NP-40, 0.5% Na deoxycholate, 1 mM EDTA, 0.1% SDS) with Protease Inhibitor Cocktail Set III, EDTA-Free diluted 1:100 (Merck Millipore, #539134), 1 mM Sodium orthovanadate, and 20 mM sodium fluoride as tyrosine and serine/threonine phosphatases inhibitors. Samples were kept on ice for 30 min and whole-cell lysates were collected after centrifugation at 13,000 rpm for 10 mins at 4 °C. Protein concentrations were determined using a Pierce BCA assay (ThermoScientific, #23228, #23224). Loading samples (4×) were prepared by boiling at 70 °C for 5 min in a Laemmli buffer (4% SDS, 20% glycerol, 10% mercaptoethanol, 0.004% bromophenol blue, and 0.125 M Tris-HCl pH 6.8). Protein samples (40 µg) were run in 10% (for AKT, pAKT, Erk1/2, pErk1/2, and VDAC1 proteins) or 12.5% (for mitochondrial electron transport system proteins) SDS-PAGE gels (Bio-Rad, USA) for 90 min at 18 mA. Samples were then transferred for 1–2 h at 120 V to 0.45 µm nitrocellulose blotting membrane (GE Healthcare, #10600002) using TE62 Transfer

tank with a cooling chamber (Hoefer™, USA). Membranes were washed 2× for 10 min with 0.05% TBST followed by blocking in 5% skimmed milk or bovine serum albumin (BSA) in 0.05% TBST for 1 h at room temperature. BSA was used for membrane blocking with phosphoprotein staining. Membranes were incubated at 4 °C overnight with the following primary antibodies diluted 1:1000 in 5% skimmed milk or BSA in 0.05% TBST solution: rabbit anti-p44/42 MAPK (Erk1/2) (Cell Signaling, #9102); rabbit anti-Phospho-p44/42 MAPK (pErk1/2) (Cell signaling, #9101); mouse anti-AKT(pan) (Cell Signaling, #9272); 1:1000 rabbit anti-Phospho-AKT(Ser473) (Cell Signaling, #9271); rabbit anti-VDAC1/Porin antibody (Abcam, #ab15895). Membranes were incubated at a 1:250 in OxPhos Rodent WB Antibody Cocktail (ThermoFisher Scientific, #458099), and rabbit anti-COX-IV monoclonal antibody (3E11) (Cell Signaling, #4850). Membranes were washed and incubated in horseradish peroxidase-(HRP) conjugated secondary antibodies, goat anti-rabbit IgG (Merck, 12–348), and goat anti-mouse IgG (Santa Cruz, #2032), diluted 1:3000 in 5% milk in 0.05% TBST. Bands were visualized using Super Signal West Femto (ThermoScientific, #34095) and developed with FUJIFILM Intelligent Darkbox Las-3000. Secondary antibody solutions conjugated to fluorescent molecules for two-color detection of AKT and pAKT proteins (anti-mouse IRDye 680 LT red and anti-rabbit IRDye 800CW green, respectively), diluted 1:8000 in 5% BSA in 0.05% TBST blocking solution for 1 h at room temperature.

Protein loading was controlled with a mouse monoclonal antibody anti-β-actin (Sigma, #A5316) for AKT and electron transport system proteins or a rabbit anti-vinculin antibody (Abcam, #155120) for Erk1/2 and VDAC1 proteins. Protein values were analyzed densitometrically with ImageJ (NIH) software. Band intensities were corrected with values determined on β-actin or vinculin blots and expressed as relative values compared to WT mice.

Detection and analysis of fluorescent bands were completed with the Li-COR Odyssey Platform (Biosciences).

### High-resolution respirometry
*Tissue sampling.* Hippocampal tissue samples were collected from WT and Tg21 animals at postnatal ages P3, 7, 14, 21, and 60. Samples were blotted dry, assessed for wet weight in a balance-controlled scale (Dual Range Analytical Balance, Mettler Toledo AG, Switzerland), maintaining constant relative humidity and hydration consistency for the stability of measures, and immediately immersed in ice-cold respiration medium MiR05 (110 mM sucrose, 0.5 mM EGTA, 3 mM MgCl₂*6H₂O, 80 mM KCl, 60 mM K-lactobionate, 20 mM taurine, 10 mM KH₂P0₄, 20 mM HEPES, and 1 g/l bovine serum albumin, pH = 7.1)[80]. All chemicals were obtained from Sigma-Aldrich (Switzerland).

*High-resolution respirometry.* Mass-specific (pmol $O_2$/s * mg tissue wet weight) OCR were collected using a high-resolution Oxygraph-2k respirometer (Oroboros, Innsbruck, Austria). Standardized instrumental calibrations were performed to correct for the back-diffusion of oxygen into the chamber from various internal components, leak from the exterior, oxygen consumption by the chemical medium, and sensor oxygen consumption. All experiments were carried out in a hyper oxygenated environment (>200 nmol/ml) to prevent any potential oxygen diffusion limitations and oxygen flux was resolved by software allowing nonlinear changes in the negative time derivative of the oxygen concentration signal (DatLab, Oroboros, Innsbruck, Austria). All measures were collected at 37 °C in a respiration buffer MiR06 (MiR05 + 280 iU/ml catalase) with saponin (50 μg/ml) to facilitate cell membrane permeabilization[81]. All substrate, uncoupler, and inhibitor titrations described below were added in series (Fig. 2a).

*Respiratory titration protocol.* The respiration protocol used examines individual aspects of respiratory control by initiating a specific sequence of respiratory states. Here we analyzed various respiratory states representative of mitochondrial proton leak (L), maximal rates of coupled oxidative phosphorylation (P), and maximal rates of uncoupled respiration (E). Each respiratory state is specific to the substrate(s), uncoupler(s), and inhibitor(s) included in the respiration medium at any given time in addition to the proton motive force and the relation of respiration to ATP production. L-state respiration, with kinetically saturating substrate and oxygen concentrations, represents uncoupled respiration due to proton leak and slippage across the inner mitochondrial membrane with a maximal proton motive force in the absence of ATP production and is comparable to the classical definition of either state 2 or 4 respiration[82]. Mitochondrially derived oxidant production is highest during L-state respiration[83,84]. P-state respiration, with kinetically saturating substrates, oxygen, and thermodynamically favorable adenylate concentrations to facilitate oxidative phosphorylation, represents respiratory rates that are well-coupled to ATP synthesis with a high proton motive force and are comparable to the classical definition of state 3 respiration[82]. E-state respiration, with kinetically saturating substrate and oxygen concentrations along with optimal exogenous protonophore provision, represents non-coupled respiratory rates independent of ATP synthesis with a collapsed proton motive force. Respiratory analysis began with the collection of L-state respiration without exogenous adenylates ($L_N$) following the addition of malate (2 mM) and octanoylcarnitine (0.2 mM). P-state respiration driven by electron input from the electron transfer flavoprotein complex and mitochondrial complex I ($P_{ETF+CI}$) was initiated with the addition of ADP (5 mM). As the

brain poorly oxidizes lipid substrates, $P_{ETF+CI}$ can serve as an internal control of sample purity. Indeed, we found negligible changes in respiration between basal, $L_N$, and $P_{ETF+CI}$ respiration (see result section). Accordingly, any reference of electron input from the electron transfer flavoprotein complex will be excluded from respiratory state identification throughout the results and discussion. P-state respiration with maximal NADH-linked electron input from mitochondrial complex I, $P_{CI}$, was induced following the additions of pyruvate (5 mM) and glutamate (10 mM). Maximal rates of P-state respiration, and the best representation of maximal oxidative phosphorylation potential in vivo, were then initiated with the addition of succinate (10 mM), which adds additional electron input from mitochondrial complex II, $P_{CI+CII}$. Maximal E-state respiration, $E_{CI+CII}$, was achieved with titrations of the protonophore, carbonyl cyanide *p*-(trifluoromethoxy) phenylhydrazone (FCCP), in 0.5 μM steps up to an optimum concentration (ranging from 1.5 to 3 μM). E-state respiration primarily reflecting electron input via mitochondrial complex II was then determined following the addition of rotenone (0.5 μM) and subsequent inhibition of mitochondrial complex I. Finally, non-mitochondrial residual oxygen consumption (ROX) was induced with the addition of antimycin A (2.5 μM) and attendant inhibition of mitochondrial complex III. All respiratory states were corrected for measures of ROX. Upon completing respiratory state analyses, ascorbate (2 mM) and $N,N,N',N'$-tetramethyl-1,4-benzenediamine, dihydrochloride (TMPD, 500 μM) were simultaneously titrated into the chambers to assess cytochrome *c* oxidase (COX; complex IV) activity via mass-specific OCR (pmol $O_2$/s * mg ww). Chemical calibrations were performed to determine and control for the auto-oxidation of TMPD occurring during measures of COX activity. For this, OCR is assessed over time using MiRO6 + saponin free of any biological sample with cytochrome *c* (10 μM), ascorbate (2 mM) and $N,N,N',N'$-tetramethyl-1,4-benzenediamine, dihydrochloride (TMPD, 500 μM) added to the chamber.

Mitochondria-specific respiration (%) was determined by normalizing mass-specific respiration (pmol $O_2$/(s*mg ww)) to COX activity (pmol $O_2$/s * mg ww); (respiratory state respiration/COX activity) * 100.

### Electron microscopy
*Perfusion and fixation.* Mice P14 were i.p. anesthetized with 40 μl pentobarbital (50 mg/ml) and, immediately following loss of pain perception, transcardially perfused with ice-cold PBS followed by 2.5%PFA/2.5%Gluteraldehyde in 150 ml sodium-cacodylate solution (0.13 M, pH = 7.4). Brains were removed from the skull and placed in a 50 ml Falcon tube containing the same fixative for post-fixation for at least 2 days, at 4 °C.

*Tissue block preparation.* 80–100 μm thick hippocampal CA1 sections were cut in a vibratome and kept overnight in Na-cacodylate buffer (0.1 M) at 4 °C.

*Osmification.* Na-cacodylate buffer was removed from the small PE tube and the tissues were incubated with 1% Osmium tetroxide (OsO₄) solution for 60 min. Tissues were then washed rapidly twice with Na-cacodylate buffer (0.1 M) and a third time for at least 30 min.

*Dehydration and embedding.* Na-cacodylate buffer was substituted with increasing concentrations of acetone: 1 × 30% acetone, 1 × 50% acetone, 2 × 70% acetone, 2 × 80% acetone, 2 × 90% acetone; each wash lasted 5 min and 3 × 100% acetone, for 5, 20 and 30 min, respectively. Tissues were washed twice with propylene oxide, 5 min per wash. Afterward, they were incubated in propylene oxide-resin I solution for 40 min at RT. Subsequently, tissues were incubated twice in resin II, which contained freshly prepared DMP30 (2,4,6-(Tri-Dimethylaminoethyl-phenol)) for 40 min, at RT.

Tissues were transferred on a glass slide to create the inclusions, covered with Aclar embedding film (ACLAR®33 C Film, 7.8 Mil, Electron Microscopy Sciences, Hatfield, PA), and incubated for 48 h, at 60 °C, for resin polymerization.

*Cutting of semithin sections.* When the resin was completely polymerized, very small regions of interest were dissected with a blade under the stereomicroscope (ZEISS, SteREO Lumar.V12) and mounted on small pyramids of resin with glue. They were left to air-dry overnight.

The mounted tissue was placed on the ultramicrotome (Leica Ultracut EM UC6, Vienna, Austria) and with a razor-sharp blade, they were given the shape of a trapezoid. The tissue was then aligned with a glass knife and semithin sections of 1-μm-thickness were cut. Sections were collected with non-magnetic tweezers and placed in small drops of ddH₂O on gelatinized glass slides. Glass slides were shortly placed on the heater and when the water drop was completely dried, then the tissue was contra stained with toluidine blue and observed with the brightfield microscope. Cutting of thin sections: Thin sections (70 nm thickness) were cut with a diamond blade and collected in ddH₂O and gathered on a nickel grid (diameter: 3.05 mm, G300-Ni, 300 lines/inch square mesh, No. 100, Science Services GmbH, Munich, Germany). The grids were put in a grid box and air-dried overnight.

*Contrasting.* A small drop of double distilled water was placed on sections, followed by drops of uranyl acetate for 7 min, washed with double distilled water; subsequent drops of citrate for 7 min; and wash with double distilled water. Samples were then left to dry.

*Image acquisition and data analysis.* Semithin sections were observed with a brightfield Axioscope 2 microscope (Carl Zeiss AG, Oberkochen, Germany) equipped with a color digital camera (AxioCam MRc5) and its corresponding software, AxioVision 4.5 (Carl Zeiss AG, Oberkochen, Germany), and then EM micrographs were taken with the 100 kV transmission electron microscope (TEM - Philips CM100 and Telos) connected to a digital CCD camera.

Mitochondria number, mitochondria size, and vesicle numbers were quantified in each field of view with automatized particle counter from Fiji ImageJ (National Institutes of Health, USA).

**Cognitive functions tests**. Morris water maze (MWM), to test hippocampal-dependent spatial learning and memory[85], novel object recognition (NOR), which combines spatial and visual memory[86], and a T-maze test for assessment of spatial working memory[87] were all performed with male and female animals of both genotypes at postnatal ages P45–60. After weaning, animals were housed in the testing room with room a 12 h light/dark inverted cycle and access to food and water ad libitum in standard laboratory conditions. Mice were controlled three times per week and handled once weekly to minimize handling-related stress that could impact performance in the task. Tests were performed during the dark phase under dimly-lit conditions (12 lux). Animals were video tracked at 4.2 Hz and 256 × 256-pixel spatial resolution using a Noldus EthoVision 1.96 system (Noldus Information Technology, Wageningen NL, www.noldus.com) throughout testing.

*Morris water maze (MWM) test (P45–50).* The MWM, test to test hippocampal-mediated spatial memory, was employed as previously described[85]. The circular black arena was 150 cm in diameter with a wall height of 50 cm. High contrast spatial cues were placed on walls of the encompassing room and inside the arena above the water surface. The arena was filled with water to a height of 15 cm and maintained at 25 °C. Day 1 consisted of teaching the mice to identify the escapable endpoint of the task by training them to find a visible Plexiglas platform (16 × 16 cm) placed randomly 0.5 cm underneath the water surface. Days 2–5 consisted of training the animals to find the hidden platform, which remained in a fixed position (acquisition phase). Throughout the acquisition phase, the water was made opaque by the addition of milk and the target platform was hidden 0.5 cm underneath the water surface 35 cm away from the surrounding walls in one quadrant (NW, NE, SE, or SW). Animals performed 6 trials of 120 s per day, with an inter-trial interval (ITI) of 30 min. On day 6, the platform was moved to the opposite quadrant (reversal phase) and four trials of 120 min were registered to measure spatial retention. The swim pathway of each mouse was automatically tracked and the time to reach the platform (escape time), total swim distance (m), and speed (m/s) were calculated.

*Novel object recognition (NOR) test (P52–55).* The NOR test relies on the animal's intrinsic preference for novelty. The choice to explore a novel object reflects learning and recognition memory, which also involves the hippocampus[88,89] together with cortical areas such as the visual cortex.

Mice were first habituated to an open field-testing arena (60 × 60 × 50 cm) with two spatial environmental cues on the walls for 10 min on two consecutive days. Mice were free to explore during habituation and the total activity time was registered. On the third day, animals were allowed to explore two equal objects placed in specific quadrants of the arena. The mice were subsequently removed from the arena for a 1-h ITI. One of the objects was replaced by another object similar in height and volume but different in shape and texture. Following this brief 1-h ITI, mice were reintroduced to the testing arena and allowed to freely explore the new object for 5 min. The next day (ITI of 24 h), mice were again allowed to explore the open testing field in the presence of a familiar and another new object. Time spent exploring each object and the number of explorations was recorded. An object discrimination ratio was calculated by (1): TN/(TN + TF). TN = time exploring the new object, TF = time exploring the familiar object. Animals that explored each object less than 10 s were excluded from the test.

*T-maze spatial working-memory test (57–60).* Spatial memory was assessed by a rewarded alternation task in a T-maze made of gray poly-vinyl-chloride/plexiglas (each arm measures 30 × 10 cm) with a removable central partition and one guillotine door for each arm of the maze. Each mouse was habituated to the T-maze for two consecutive days before testing by allowing two mice to explore the T-maze with a food reward (pellets). The test was performed with 6 trials over two consecutive days. One trial consisted of two successive runs through the maze. In the first run (forced run) a food reward (R) was placed in an open arm. The other arm was closed with a barrier. Mice were then immediately returned to the start arm for 5 s and allowed a second run in which both arms were available, so the animals could enter either arm (choice run), however, food was available only in the arm closed on the first run. Consequently, spatial working memory is necessary to remember which arm was open in the first trial and to alternate. A run was terminated if a choice was not made within 2 min. The sample arm available to the mice on the first forced run varied randomly from trial to trial. Following each trial, the mice were removed from the T-maze and returned to their cage for a 1-h ITI (short-term memory). This spatial working-memory task is quantified by spontaneous alternation. The percentage of alternation and time to reach the reward was calculated per animal.

**Statistics and reproducibility**. For all statistical evaluations included in this study, unless otherwise specified, an α of $p < 0.05$ was considered significant and data are reported as mean ± SD.

Differences between genotypes (WT vs. Tg21) across ages in protein expression, mitochondrial respiration, and qPCR (Fig. 1, Supplementary Fig. 1b, d; Fig. 2b–d, Supplementary Fig. 2, Figs. 3a–d, 4, 5k–l, 6a–f) were determined using a 2-way ANOVA and post hoc analyses identifying individual group differences were conducted with Sidak's correction for multiple comparisons.

Main effects of genotype (WT vs. Tg21) on spleen weight, hematocrit, hemoglobin, plasma EPO (Supplementary Fig. 1a); mitochondria number (Fig. 6h), mitochondria size (Fig. 6i), mitochondrial mass in presynaptic terminals (Fig. 6j) and vesicle numbers (Fig. 5k); were determined using unpaired Student's t-test.

Comparisons of rhEPO expression in brain, kidney, liver, and spleen in WT and Tg21 mice (Supplementary Fig. 1c), and the general differences across respiratory states in WT (Figs. 2e, 3e and Supplementary Fig. 2c, d) and Tg21 mice (Figs. 2f, 3f and Supplementary Fig. 2c, d), were analyzed with a one-way repeated measures ANOVA and individual differences identified using Tukey's HSD post hoc test.

Effects of EPO (i.p.) injections on hematocrit (Fig. 5b), protein expression: AKT (Fig. 5d), Erk1/2 (Fig. 5e), VDAC1 (Fig. 5f), pAKT (Fig. 5h), pERK1/2 (Fig. 5i); and COX activity (Fig. 5j) were determined using unpaired Student's t-test.

The main effects of genotype for outcome variables derived from the behavioral tests were analyzed using a 2-way ANOVA and post hoc analyses identifying individual group differences were conducted with Sidak's correction for multiple comparisons (GraphPad Prism 8.0.1.).

All tests were performed using Prism 8.0.1. GraphPad Software, San Diego, CA, USA.

**Reporting summary**. Further information on research design is available in the Nature Research Reporting Summary linked to this article.

## Data availability
The data that support the findings of this study are available within the paper and its supplementary information files. Source data underlying figures are presented in Supplementary Data 1. Individual data sets generated during the cognitive tests are available from the corresponding author on reasonable request.

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

## Acknowledgements

This work was supported by the University of Zürich, the Baugenossenschaft Zurlinden and the Swiss National Science Foundation by the MHV Grant PMPDP3_145480 to E.M. S.G. J.S. is supported by the "Fonds de recherché du Quebec-Santé" (FRQ-S; FQ121919). We kindly acknowledge Prof. Jean-Marc Fritschy for his great scientific feedback and for proofreading of the manuscript. Many thanks to Nicole Kachappilly, Dr. Tatjana Haenggi, and Cornelia Schwerdel for their excellent technical support.

## Author contributions

E.M.S.G. designed the study; R.A.J., M.A.A., C.K.-H., P.M., C.A.-R., S.L. and E.M.S.G. performed research and data analyses; M.T., R.A.J. and J.S. provided expertise with mitochondria measurements and intellectual input; E.M.S.G. and R.A.J. wrote the manuscript; M.G. provided the Tg21 mice, his expertise, and supported to conduct this project; and E.M.G.S., R.A.J., M.A.A., C.A.-R., S.L., C.K.-H., M.T., J.S. and M.G. proofread the manuscript.

## Competing interests

The authors declare no competing interests.
