## [Peer Review File · Communications Biology]

Reviewers' comments:

Reviewer #1 (Remarks to the Author):

The authors of this manuscript investigated effects of erythropoietin overproduction on brain function, using transgenic mice (Tg21) expressing human erythropoietin in brain exclusively. The mice showed that erythropoietin overload in neonatal brain may affect brain function such as spatial memory of mice through increasing mitochondrial content in pyramidal neurons. However, no approach to elucidation of the association between the increased mitochondria and brain function was conducted. Additionally, their conclusion is based on data from only one line of transgenic mice under extraordinary conditions.

Critical concerns

Because all data in this manuscript are obtained by using one line of transgenic mice expressing human erythropoietin at high levels in brain, other approaches must be tested to confirm their conclusion. For example, intracerebral injection of erythropoietin into wild-type mice and intracerebral injection of erythropoietin-neutralizing antibody into Tg21. Additionally, the possibility of transgene integration effect on the brain phenotype should be excluded by appropriate experiments.

As well as the Erk-AKT pathway, the JAK-STAT pathway is known as the intracellular pathways stimulated by erythropoietin in erythroid cells, and significance of the JAK-STAT pathway for neural erythropoietin signaling has been reported. However, this study analyzed only the Erk-AKT pathway. The authors must not ignore the JAK-STAT pathway.

The erythropoietin receptor (EPOR) expression was investigated by means of RT-PCR and in situ hybridization in the brain. The expression levels should be compared to those in erythroid cells as their positive controls.

The authors insist that the mitochondrial content in the hippocampus of the erythropoietin-overexpressing mice is increased compared to that of wild-type mice. The size and shape of mitochondria in the transgenic hippocampus must be examined by means of electron microscopy.

The discussion section is long-winded, containing points not worth considering in this manuscript.

To describe "These findings highlight the clinical potential of EPO to maintain healthy hippocampal maturation and prevent neurodevelopmental delay after perinatal hypoxic-ischaemic injury (in Conclusion)", approaches other than Tg21 mice are essential.

Reviewer #2 (Remarks to the Author):

The authors show that overexpressing EPO in TG21 mice increases mitochondrial biogenesis in the hippocampus and improves cognition in mice during young adulthood (P60). They report that EPO acts by increasing p-Erk1/2, p-Akt.

These findings add to growing evidence that supports a role for mitochondria in neuroprotection and cognition.

While the conclusions are sound for the most part, some clarification would enhance the study.

Studies have also shown that EPO rescues cognition in mouse models of Alzheimer's disease. These references should be included.

In Fig. 1g the FISH is difficult to see. There should be a higher magnification image. Also, what cell types was EPOR expressed in?

Mitochondrial respiration was normalized to COX activity to determine that respiration was enhanced by an increase in mitochondrial mass or biogenesis. Their conclusion that EPO increases mitochondrial biogenesis could be strengthened by additional experiments.

Are any mitochondrial biogenesis factors or transcriptional regulators increased? PPARGC1a,TFAM?
Is the ratio of mitochondrial DNA/genomic DNA increased?

In respirometry experiments, since tissue was analyzed, changes in neurons could be masked by mitochondria in glial cell types. Fluorocitrate could be added to inhibit glial respiration in order to examine neuronal mitochondria.

In Fig. 5 Western blots, why were mitochondrial electron transport complex proteins normalized to actin? These should be quantitated by Western blotting with with mitochondrial fractions instead of cytoplasmic fractions.

The b-actin also indicates the loading isn't very consistent.

Also in Fig. 5, why is complex I increased in WT at P60, but very low in TG21 at P60? This isn't reflected in the quantitation in Fig. 5d.

Fig. 4c and d are difficult to interpret. This needs some clarification.

Minor points:

EPO has been shown to increase neuronal hemoglobin. Is neuronal hemoglobin increased in the hippocampus in Tg21 mice?

There are some English language mistakes.

Reviewer #3 (Remarks to the Author):

In the manuscript "Erythropoietin enhances postnatal hippocampal mitochondrial content, function, and cognition", Jacobs et al, using transgenic animals, investigated the effect of brain tissue-specific EPO overexpression on brain functions and mitochondria energetics during neonatal and adolescent period in rats. The brain overexpression of EPO on signaling pathways, like Act and ERK, within the brain at different postnatal time points was addressed. In addition, authors perform a series of important controls, showing that brain-specific EPO overexpression does not affect hematocrit and, in general, the local EPOR expression is not altered at different post-natal time periods. These finding, particularly for the developing brain, is novel and important for potential clinical application of rhEPO for neonates.

The described methods and statistical analysis are detailed and clearly written.

Major comments:

1) Authors claims in the beginning of the discussion that "... EPO overexpression primarily influences hippocampal postnatal development.", however, the development per se is not addressed in the manuscript. For example, the number or ratio of mature vs immature neurons or other neuronal cell populations, like oligodendrocytes at the different maturation stages are not investigated. On page 15 of the Discussion section, authors claims "Increased oxidant strain would have ultimately been detrimental to neural maturation and hippocampal function, neither of which were observed in the current study." - how neuronal maturation was actually determined? Authors speculate in the discussion (page 19), that "... EPO induced improved mitochondria number and function in the brain improves performance time and learning,... most likely as an influence on synaptic number and plasticity into early adulthood" – however no direct or indirect approaches to quantify the synaptic density was included in the study.

The comparative analysis of EPOR expression and ERK/Act phosphorylation across different ages was done. This information should be linked to developmental parameters of hippocampus formation, then authors may say that EPO overexpression is affect hippocampal postnatal development. Alternatively, authors should carefully re-consider the discussion regarding the hippocampal maturation in response to EPO overexpression.

2) Overexpression of EPO has an effect on memory; this was shown in Morris Water Maze and novel object recognition tests, but has no effect on memory evaluated with T-maze test. Authors explain the absence of effect on memory in T-maze by claiming that hippocampus is not involved in T-maze based memory formation. Few examples: "...T-maze testing of working memory, which is a complex function that involves storage, organization, and update of information and involves the prefrontal cortex" and "...T-maze working memory, which involves areas of the brain other than the hippocampus...". This is in contrast to the classical view that hippocampus is involved in formation of working memory particularly in the spatial memory evaluated by T-maze. For example, see a review: "Spatial memory: Theoretical basis and comparative review on experimental methods in rodents (Behav Brain Res. 2009). This point should be very carefully considered by authors.

Minor comments:

An overview figure showing the time line of in vivo behavior tests and at what time the brain samples were collected – would improve the manuscript.

Page 7: The abbreviation OCR, FCCP and ROX are mentioned for the first time without opening that – please correct. The same for COX at page 8 and ITIs and ODI at page 12.

We extend our sincere appreciation to the associate editor and all three reviewers for their time and thoughtful feedback regarding our submitted research. We considered all input thoroughly and addressed as many issues as completely as was possible. We believe that our revised work reflects the collective efforts achieved through this successful review process and is now a more complete and better supported study. Again, we thank everyone for their contribution and hope you find that our guided efforts now reflect your thoughtful contributions to our work.

Associate Editor, Dr. George Inglis (Remarks in the first decision):

In particular, we agree with the reviewers that key revisions would include:

(1) Further assessing altered mitochondrial biogenesis in tg animals (as suggested by Reviewer 2, perhaps through mtDNA copy number assays or expression of key mitochondrial transcription factors).

These are very appropriate suggestions that we agree would more completely explain our findings. Accordingly, we have assessed mitochondrial to nuclear DNA (mtDNA / nDNA) ratio in wild type (WT) and transgenic mice overexpressing cerebral EPO (Tg21), which is now included as panel (a) of Figure 4. These additional measures support our previous findings to suggest upregulated hippocampal mitochondria in postnatal (P) ages of 14 and 21.

We also assessed PPARGC1A gene expression at P3, P7, P14, and P21 in both genotypes, which is presented in panel (a) of Figure 6. PGC1 α mRNA increases in an age-dependent manner (P3 & P7 < P14 & P21) in both genotypes and was identified as higher in Tg21 vs WT animals at P21.

(2) Examining altered neuronal morphology and/or development in aging tg mice

Although we agree that examining altered neuronal morphology and/or development in aging Tg21 mice would improve the current study, the limitation in time and resources did not allow us to completely assess these variables throughout development. In attempt to satisfy all suggestions as completely as possible, we have, however, assessed mitochondrial size and number in hippocampal tissue in P14 mice of both genotypes. We now report that mitochondrial number (Figure 6h) and mitochondrial size (Figure 6i) were both greater in Tg21 vs WT. To better assess morphological differences across genotypes, we examined mitochondria and vesicle numbers specifically in presynaptic hippocampal neurons. These morphological assessments identified greater mitochondrial mass (Figure 6j) and vesicle number (Figure 6k) in Tg21 vs WT groups. Representative images of our EM assessments are also presented in Figure 6g, illustrating the differences in mitochondria and presynaptic vesicle numbers between genotypes. We hope the editor and reviewers find these additions as beneficial efforts to address altered neuronal morphological assessments across genotypes. Also, we have recently published additional findings in line with these suggestions (<https://pubmed.ncbi.nlm.nih.gov/33495244/>), which demonstrates that EPO stimulates postnatal GABAergic maturation in the hippocampus. We show an increase in

hippocampal GABA-immunoreactive neurons, and postnatal elevation of interneurons expressing parvalbumin (PV), somatostatin (SST), and neuropeptide Y (NPY). Analysis of perineuronal net (PNN) formation and innervation of glutamatergic terminals onto PV+ cells, shows to be enhanced early in postnatal development. Additionally, an increase in GABAergic synapse density and IPSCs in CA1 pyramidal cells from Tg21 mice is observed. Detection of EPO receptor (EPOR) mRNA was observed to be restricted to glutamatergic pyramidal cells and increased in Tg21 mice at postnatal day (P)7, along with reduced apoptosis. Our findings show that EPO can stimulate postnatal GABAergic maturation in the hippocampus, by increasing neuronal survival, modulating critical plasticity periods, and increasing synaptic transmission. We believe that the combination of our additional experiments along with relevant research we have since published addresses these concerns.

(3) Adding an additional control to demonstrate the relevance of EPO toward observed phenotypes (perhaps supplementing WT mice with exogenous EPO, as suggested by Reviewer 1).

We are happy to report that we were able to add to our initial findings by including an additional set of experiments completed on recently weaned mice (P26) following three consecutive days of intraperitoneal injections of EPO high enough to facilitate transport across the blood-brain barrier vs saline treated animals. The results from that set of experiments are presented in Figure 5. High-doses of systemic EPO were shown to result in canonical adaptations, such as increased haematocrit (Hct; Figure 5b), as well as increased AKT and ERK 1/2 activation (Figures 5h and 5i, respectively), increased voltage-dependent anion-selective channel 1 (VDAC1; Figure 5f), increased cytochrome c oxidase (COX) activity (Figure 5j) and mass-specific rates of hippocampal respiration (Figure 5k). Mitochondria-specific respiration (mass-specific respiration normalized to COX activity) was not influenced by systemic EPO treatments. These control experiments evidence additional support demonstrating the relevance of EPO directing phenotypes in young mice and support our initial findings in Tg21 v WT animals.

Reviewer #1 (Remarks to the Author):

The authors of this manuscript investigated effects of erythropoietin overproduction on brain function, using transgenic mice (Tg21) expressing human erythropoietin in brain exclusively. The mice showed that erythropoietin overload in neonatal brain may affect brain function such as spatial memory of mice through increasing mitochondrial content in pyramidal neurons. However, no approach to elucidation of the association between the increased mitochondria and brain function was conducted. Additionally, their conclusion is based on data from only one line of transgenic mice under extraordinary conditions.

We extend our sincere gratitude to Reviewer 1 for their time and thoughtful critiques of our research. We have worked to satisfy your concerns as best we could and hope that our resubmission reflects your thoughtful guidance of our continued efforts.

Critical concerns

Because all data in this manuscript are obtained by using one line of transgenic mice expressing human erythropoietin at high levels in brain, other approaches must be tested to confirm their conclusion. For example, intracerebral injection of erythropoietin into wild-type mice and intracerebral injection of erythropoietin-neutralizing antibody into Tg21. Additionally, the possibility of transgene integration effect on the brain phenotype should be excluded by appropriate experiments.

We appreciate these valid concerns. As detailed above in our response to the associate editor, we have included a set of control experiments in which P23 mice received 3 consecutive days of high-dose i.p. EPO in attempt to better control for our experiments using Tg21 mice.

As well as the Erk-AKT pathway, the JAK-STAT pathway is known as the intracellular pathways stimulated by erythropoietin in erythroid cells, and significance of the JAK-STAT pathway for neural erythropoietin signaling has been reported. However, this study analyzed only the Erk-AKT pathway. The authors must not ignore the JAK-STAT pathway.

We understand the reviewer's general concern regarding assessment of intracellular JAK-STAT pathway control. We also expect the JAK-STAT pathway to be stimulated by EPO. Unfortunately, detection of STAT5 phosphorylation was unspecific in our WB experiments. Since, AKT and Erk1/2 pathways are more commonly reported as key regulating pathways in mitochondrial biogenesis and function, we considered those pathways essential for our study.

The erythropoietin receptor (EPOR) expression was investigated by means of RT-PCR and in situ hybridization in the brain. The expression levels should be compared to those in erythroid cells as their positive controls.

Thank you for this suggestion. Our Tg model overexpresses EPO only in the brain (Fig. 1a), with no observed systemic changes (Fig. S1c). Therefore, we are interested in a relative analysis of brain EPOR expression between control and Tg mice and not in a quantitative analysis correlated to erythroid cells.

The authors insist that the mitochondrial content in the hippocampus of the erythropoietin-overexpressing mice is increased compared to that of wild-type mice. The size and shape of mitochondria in the transgenic hippocampus must be examined by means of electron microscopy.

Again, thank you for this suggestion. We have assessed mitochondrial number and size via electron microscopy in P14 Tg21 and WT animals. Those results are presented in Figure 6.

The discussion section is long-winded, containing points not worth considering in this manuscript.

We apologize for our initial oversight on the brevity of our discussion. We have revised the discussion and, accordingly, have eliminated discussion on topics that we thought could be removed. We hope our efforts to alleviate this concern are apparent and improve the quality of the manuscript.

To describe “These findings highlight the clinical potential of EPO to maintain healthy hippocampal maturation and prevent neurodevelopmental delay after perinatal hypoxic-ischaemic injury (in Conclusion)”, approaches other than Tg21 mice are essential.

We have revised this final sentence in the Conclusion to now read, “*These findings highlight a therapeutical potential of EPO to stimulate hippocampal metabolism during maturation, which may improve neurodevelopment and cognition...*” We hope that this better reflects a conclusion that is more appropriately supported by our data.

Reviewer #2 (Remarks to the Author):

The authors show that overexpressing EPO in TG21 mice increases mitochondrial biogenesis in the hippocampus and improves cognition in mice during young adulthood (P60). They report that EPO acts by increasing p-Erk1/2, p-Akt. These findings add to growing evidence that supports a role for mitochondria in neuroprotection and cognition. While the conclusions are sound for the most part, some clarification would enhance the study.

We extend our sincere gratitude to Reviewer 2 for their time and thoughtful critiques of our research. We have worked to satisfy your concerns as best we could and hope that our resubmission reflects your thoughtful guidance of our continued efforts.

Studies have also shown that EPO rescues cognition in mouse models of Alzheimer's disease. These references should be included.

We thank you for pointing this out and have included reference citations for published research demonstrating that EPO rescues cognition in mouse models of Alzheimer's disease in Sun, J. et al., 2019; Dmytriyeva, O. et al. 2019; Esmaeili T. et al., 2015; Jarero-Basulto, et. al, 2020; Rey F. et al, 2019 and 2021; Urena-Guerrero, et al., 2020; Ehrenreich, et al., 2008.

In Fig. 1g the FISH is difficult to see. There should be a higher magnification image. Also, what cell types was EPOR expressed in?

Again, we thank you for bringing this to our attention. We removed our FISH analysis from the manuscript since we recently demonstrated that EPO functions on CA1 pyramidal cells in the hippocampus and facilitates hippocampal postnatal development by reducing neuronal cell death and promoting synaptogenesis (<https://pubmed.ncbi.nlm.nih.gov/33495244/>). This reference has been included to specify this important point in the introduction of the revised manuscript.

Mitochondrial respiration was normalized to COX activity to determine that respiration was enhanced by an increase in mitochondrial mass or biogenesis. Their conclusion that EPO increases mitochondrial biogenesis could be strengthened by additional experiments. Are any mitochondrial biogenesis factors or transcriptional regulators increased? PPARGC1a, TFAM? Is the ratio of mitochondrial DNA/genomic DNA increased?

We agree with you and appreciate the benefit that the additional empirical scrutiny would provide our initial work. Accordingly, we have assessed genetic expression of PPARGC1a (higher in Tg21 v WT mice at P21), quantification of mitochondrial number and size via electron microscopy (both higher in Tg21 v WT mice), and specifically assessed the ratio of mtDNA to nDNA at P7, P14, and P21 (higher ratio in Tg21 v WT mice at P14 and P21), as suggested. We believe, and hope you agree, that this evidence more completely supports our claims of a more robust drive for mitochondrial biogenesis in Tg21 mice hippocampus during postnatal development.

In respirometry experiments, since tissue was analyzed, changes in neurons could be masked by mitochondria in glial cell types. Fluorocitrate could be added to inhibit glial respiration in order to examine neuronal mitochondria.

We greatly appreciate the suggestion to utilize fluorocitrate during respiratory measures to specify glial v neuronal respiratory contributions to overall hippocampal oxidative potential, and we are very interested in utilizing this control in future experiments.

In our attempt to satisfy your legitimate concerns regarding the hippocampal specificity of mitochondrial increases/expansion, we have assessed appropriate mitochondrial characteristics via electron microscopy (Figure 6). These assessments provide evidence to support neuronal-specific increases in mitochondria, hopefully alleviating some concern.

In Fig. 5 Western blots, why were mitochondrial electron transport complex proteins normalized to actin? These should be quantitated by Western blotting with mitochondrial fractions instead of cytoplasmic fractions. The b-actin also indicates the loading isn't very consistent. Also in Fig. 5, why is complex I increased in WT at P60, but very low in TG21 at P60? This isn't reflected in the quantitation in Fig. 5d.

Thank you for directing us to these concerns. We normalized the OXPHOS proteins to b-actin because the protein extraction protocol for OXPHOS proteins is different than for VDAC1 and could not be identified in the same blot. Given the extent of mitochondrial data collected (high-resolution respirometry, electron microscopy, mtDNA to nDNA ratio, cytochrome C oxidase activity, and the genetic expression of PPARGC1a), we are confident that the results of our Western blots are not spurious, but rather reflect and support the collective mitochondrial assessments.

Also in Fig. 5, why is complex I increased in WT at P60, but very low in TG21 at P60? This isn't reflected in the quantitation in Fig. 5d.

We thank you for challenging us to include a more representative gel sample in relation to our collective findings and have provided a more accurate gel sample for our representative image. Complex I protein increases a little more in Tg21 mice than in WT mice on day 60, and quantification of relative CI protein expression demonstrates that Tg21 are higher than in WT but without reaching statistical significance.

Fig. 4c and d are difficult to interpret. This needs some clarification.

We apologize for an incomplete/inadequate initial description of this data and have removed this set of data for simplification.

Minor points:

EPO has been shown to increase neuronal hemoglobin. Is neuronal hemoglobin increased in the hippocampus in Tg21 mice?

We thank you for the inquiry. We remain unfamiliar with neuronal hemoglobin expression in our model, but since EPO treatment increases brain hemoglobin expression and neuronal mitochondrial activity in mouse models of multiple sclerosis (MS) (Singhal, N., et al. 2018), it is possible that EPO also causes an increase in brain hemoglobin during development stimulating neuronal mitochondrial function. We have included this insight to our discussion. This is a very interesting question to address in future research.

There are some English language mistakes.

Again, we apologize for grammatical mistakes included in our initial submission. We have worked to improve our efforts to submit a manuscript using clear and accurate English.

Reviewer #3 (Remarks to the Author):

In the manuscript "Erythropoietin enhances postnatal hippocampal mitochondrial content, function, and cognition", Jacobs et al, using transgenic animals, investigated the effect of brain tissue-specific EPO overexpression on brain functions and mitochondria energetics during neonatal and adolescent period in rats. The brain overexpression of EPO on signaling pathways, like Act and ERK, within the brain at different postnatal time points was addressed. In addition, authors perform a series of important controls, showing that brain-specific EPO overexpression does not affect hematocrit and, in general, the local EPOR expression is not altered at different postnatal time periods. These finding, particularly for the developing brain, is novel and important for potential clinical application of rhEPO for neonates. The described methods and statistical analysis are detailed and clearly written.

We extend our sincere gratitude to Reviewer 3 for their time and thoughtful critiques of our research. We have worked to satisfy your concerns as best we could and hope that our resubmission reflects your thoughtful guidance of our continued efforts.

Major comments:

1) Authors claims in the beginning of the discussion that "... EPO overexpression primarily influences hippocampal postnatal development.", however, the development per se is not addressed in the manuscript. For example, the number or ratio of mature vs immature neurons or other neuronal cell populations, like oligodendrocytes at the different maturation stages are not investigated. On page 15 of the Discussion section, authors claims "Increased oxidant strain would have ultimately been detrimental to neural maturation and hippocampal function, neither of which were observed in the current study." - how neuronal maturation was determined? Authors speculate in the discussion (page 19), that " ... EPO induced improved mitochondria number and function in the brain improves performance time and learning,... most likely as an influence on synaptic number and plasticity into early adulthood" – however no direct or indirect approaches to quantify the synaptic density was included in the study.

These are important points to clarify, and we thank you for directing us to these initial points of discussion. We have since published some of the major changes caused by EPO overexpression to hippocampal neurodevelopment including neuronal density and synaptogenesis (Khalid, et. al, 2021), and we have referenced these findings the revised manuscript. Additionally, we provide EM analysis to support the increase in mitochondria numbers in neurons and axon presynaptic terminals.

The comparative analysis of EPOR expression and ERK/Act phosphorylation across different ages was done. This information should be linked to developmental parameters of hippocampus formation, then authors may say that EPO overexpression is affect hippocampal postnatal development. Alternatively, authors should carefully re-consider the discussion regarding the hippocampal maturation in response to EPO overexpression.

Again, we thank you for this salient point and helpful suggestion. Accordingly, we have revised the entire discussion and made a point to better identify the roles of ERK/AKT pathways to developmental parameters of hippocampus formation in the section entitled, '*EPO/EPOR signalling in the hippocampus*'.

2) Overexpression of EPO has an effect on memory; this was shown in Morris Water Maze and novel object recognition tests, but has no effect on memory evaluated with T-maze test. Authors explain the absence of effect on memory in T-maze by claiming that hippocampus is not involved in T-maze based memory formation. Few examples: "...T-maze testing of working memory, which is a complex function that involves storage, organization, and update of information and involves the prefrontal cortex" and "...T-maze working memory, which involves areas of the brain other than the hippocampus...". This is in contrast to the classical view that hippocampus is involved in formation of working memory particularly in the spatial memory evaluated by T-maze. For example, see a review: "Spatial memory: Theoretical basis and comparative review on experimental methods in rodents (Behav Brain Res. 2009). This point should be very carefully considered by authors.

Given your eloquent critique, we have carefully reworded our interpretation specific to hippocampal dependence of memory formation in our tests. Our T-maze tests was designed to include spatial memory. We indented to intially convey that “*T-maze working memory, which involves other brain areas including the hippocampus...*”. We also referenced the protocol we utilized (Wenk, G.L., 2001).

Minor comments:

An overview figure showing the time line of in vivo behavior tests and at what time the brain samples were collected – would improve the manuscript.

As suggested, we have included an overview figure showing the timeline when brain samples were collected, and behavioral tests conducted (Fig. 1a). We thank the reviewer for this suggestion and hope that the overview helps to improve the clarity in our revised manuscript.

Page 7: The abbreviation OCR, FCCP and ROX are mentioned for the first time without opening that – please correct. The same for COX at page 8 and ITIs and ODI at page 12.

Thank you for noting the discrepancies in first-used abbreviations and their definitions. Our overall manuscript format had shifted the methods section from second to last and the necessary definition of first-used abbreviations was missed. We apologize for this oversite and have adjusted the manuscript to appropriately define first-used abbreviations.

REVIEWERS' COMMENTS:

Reviewer #1 (Remarks to the Author):

The authors addressed most of the concerns raised by the reviewers, but there are 2 concerns remaining.

Original comment:

As well as the Erk-AKT pathway, the JAK-STAT pathway is known as the intracellular pathways stimulated by erythropoietin in erythroid cells, and significance of the JAK-STAT pathway for neural erythropoietin signaling has been reported. However, this study analyzed only the Erk-AKT pathway. The authors must not ignore the JAK-STAT pathway.

Response:

We understand the reviewer's general concern regarding assessment of intracellular JAK-STAT pathway control. We also expect the JAK-STAT pathway to be stimulated by EPO. Unfortunately, detection of STAT5 phosphorylation was unspecific in our WB experiments. Since, AKT and Erk1/2 pathways are more commonly reported as key regulating pathways in mitochondrial biogenesis and function, we considered those pathways essential for our study.

New comment:

For general readers, the authors are strongly recommended to explain that the data of STAT phosphorylation are omitted in this paper because the AKT/Erk pathways are more commonly involved in mitochondrial biogenesis compared to the other EPO downstream pathways, including the JAK-STAT pathway. Just for information to the authors, STAT3 is dominantly activated by neural EPO signalling rather than STAT5.

Original comment:

The erythropoietin receptor (EPOR) expression was investigated by means of RT-PCR and in situ hybridization in the brain. The expression levels should be compared to those in erythroid cells as their positive controls.

Response:

Thank you for this suggestion. Our Tg model overexpresses EPO only in the brain (Fig. 1a), with no observed systemic changes (Fig. S1c). Therefore, we are interested in a relative analysis of brain EPOR expression between control and Tg mice and not in a quantitative analysis correlated to erythroid cells.

New comment:

Since expression levels of EPOR are closely related to cellular sensitivity to EPO, comparing the EPOR expression levels of the hippocampus (Figure 1c) to those of erythroid cells (or spleens) will provide important information. If the levels are low, high concentrations of EPO in the brain, which are derived from EPO transgene or EPO injection, are suggested to be essential for promoting hippocampal mitochondrial function in mice.

Reviewer #2 (Remarks to the Author):

The authors have addressed my concerns. Additional data strengthen their conclusion that EPO stimulates mitochondrial biogenesis during development of hippocampal neurons that then supports cognition in early adulthood.

Reviewer #3 (Remarks to the Author):

The authors have addressed all my comments. I recommend for publication.

Minor point: In the table 1 authors indicated GAPDH and ACTB primers, but didn't mentioned how they were used. Actually these genes were not mentioned in the text of the manuscript at all. Please correct.